# Brochosomes as an antireflective camouflage coating for leafhoppers

Wei Wu[1]*, Qianzhuo Mao[1], Zhuang-Xin Ye[1], Zhenfeng Liao[2], Hong-Wei Shan[1], Jun-Min Li[1], Chuan-Xi Zhang[1], Jian-Ping Chen[1]*

[1]State Key Laboratory of Agricultural Products Safety, Key Laboratory of Biotechnology in Plant Protection of MARA, Zhejiang Key Laboratory of Green Plant Protection, Institute of Plant Virology, Ningbo University, Ningbo, China; [2]State Key Laboratory for Sustainable Control of Pest and Disease, Institute of Virology and Biotechnology, Zhejiang Academy of Agricultural Sciences, Hangzhou, China

## eLife Assessment

The authors provide **important** insights into a system of insect camouflage where a coating of self-made nanoparticles (brochosomes) reduces the reflection of UV light, leading to lower predation by spiders. **Compelling** evidence is provided by micro-UV-Vis spectroscopy, electron microscopy, transcriptome and proteome analysis, histology, in vivo predation assays, and gene knockdowns. The phylogenetic analyses provide evidence that the genes coding for the brochosome proteins are clade-specific and have diversified by gene duplication.

*For correspondence:
wuwei_19861115@163.com (WW);
jianpingchen@nbu.edu.cn (J-PC)

**Competing interest:** The authors declare that no competing interests exist.

## Abstract

In nature, insects face immense predation pressure, where visual cues play a vital role in predators locating them. To counter this threat, insects employ a variety of nano- and microstructures on their cuticular layer to manipulate and interact with light, enhancing antireflective properties and providing camouflage or reducing detectability by predators. Leafhoppers have a unique extracuticular coating called brochosome, yet its antireflective functions and protein composition remain unclear. Our study demonstrates strong antireflective properties of brochosomes, effectively reducing reflectance on the cuticle surface, especially in the ultraviolet spectrum, to improve evasion from visual predators. Furthermore, we identify four novel structural proteins of the brochosome (BSM) for the first time. Inhibiting their synthesis by RNAi alters brochosome morphology, impacting the optical properties of the cuticle surface. Evolutionary origin analysis of BSM suggests that brochosomes likely originated from a process involving duplication–divergence. Our study reveals that leafhoppers employ a unique camouflage strategy by secreting brochosomes as antireflection nanocoatings, enabling them to evade natural predators and contributing to their evolutionary success.

## Introduction

Predation exerts significant selective pressure on the evolution of various species, driving the development of antipredator strategies across the animal kingdom (*Galloway et al., 2020*; *Stevens and Ruxton, 2019*). Camouflage is a widely employed strategy, allowing animals to blend into their surroundings by adjusting their coloration and patterns to reduce detection by visual predators (*Hughes et al., 2019*; *Skelhorn and Rowe, 2016*). Nevertheless, the presence of predators with multispectral vision, capable of perceiving nonvisible light spectra like infrared and ultraviolet (UV), can compromise traditional color and pattern-based camouflage (*Galloway et al., 2020*; *Jones et al., 2007*; *Stevens, 2007*; *Thery and Gomez, 2010*), paralleling the challenges of military camouflage

**eLife digest** Animals have evolved various strategies to hide from or escape predators, one of which is camouflage. This adaptation helps animals blend into their surroundings using specific colors or patterns. However, many predators possess multispectral vision, enabling them to detect different wavelengths of light, which can make camouflage based on visible light ineffective.

Insects face substantial predation pressure from visual predators such as birds, reptiles and other arthropods. Many insects dwell in environments with low light reflectivity, such as leaves, tree bark, or soil, and any light reflection from their bodies could increase their visibility.

In response to these selective pressures, many insects have evolved specialized antireflective structures in their cuticle, wings and eyes. For example, leafhoppers have a unique coating on their cuticle known as the brochosome. Brochosomes are hollow, honeycomb-like spheres with a diameter ranging from 0.2 to 0.6 micrometers. They are produced in the excretory system called the Malpighian tubules and are secreted through the hindgut, where they are applied as a coating to the new cuticle after molting.

Brochosomes consist of lipids and proteins, but their exact composition and role remain unclear. Some researchers believe that brochosomes may serve as a protective barrier or antireflective layer. To investigate the camouflage role of brochosomes, Wu et al. used a combination of imaging and gene and protein analyses to study the brochosome coverage and UV reflectance in 5 to 25-day-old male and female leafhoppers of the Cicadellidae family.

The results indicated that older individuals had fewer brochosomes than younger ones. The experiments also revealed that brochosomes significantly reduced the reflection of ultraviolet light from the surface of leafhoppers by around 30% and diminished the reflectance of visible and infrared light.

Next, Wu et al. conducted predation experiments using jumping spiders known to prey on leafhoppers by analyzing the time of the first attack and the feeding behavior of the spider. The findings showed that jumping spiders preferred to attack older individuals with less brochosome coverage. This suggests that brochosomes are particularly beneficial for younger leafhoppers.

Gene and protein analyses identified four structural brochosome proteins. Experimentally blocking these proteins induced changes to the morphology of the brochosomes and modified the characteristics of the leafhoppers' cuticle, including an increased diameter and structural deformation of the honeycomb architecture, alongside elevated UV reflectance. Phylogenetic analysis of the corresponding genes revealed that these proteins likely evolved through gene duplication events followed by a gradual accumulation of genetic modifications.

The study of Wu et al. demonstrates that brochosomes serve as an antireflective camouflage coating in leafhoppers to evade visual predators. The unique structure and composition of the lipids and proteins making up the brochosomes appear to be responsible for the antireflective properties of this coating. Further studies will advance our understanding of insect antipredator adaptations and evolutionary mechanisms underlying ecological niche specialization. Furthermore, they may provide biomimetic insights relevant for developing advanced camouflage technologies by emulating biological nanostructures such as brochosomes.

---

under nonvisible light (*Xi et al., 2023*). Consequently, achieving alignment between an animal's surface optical characteristics and its surroundings becomes a crucial requirement in visual camouflage strategies.

Insects, one of the most diverse animal groups on earth, contend with intense predation pressure (*Cinel et al., 2020*; *Sheikh et al., 2017*). Many of their predators, including birds, reptiles, and predatory arthropods, rely on visual cues, with some possessing highly developed UV vision, where UV light plays a critical role in locating insect prey (*Lim and Ben-Yakir, 2020*; *Stobbe et al., 2009*). Furthermore, the natural backgrounds where insects reside, such as leaves, tree bark, and soil, exhibit minimal light reflection (*Endler, 1993*). Consequently, light reflections originating from an insect's body significantly elevate its risk of exposure to predators (*Tovee, 1995*). To counter this challenge, the various nano- and microstructural features of the insect cuticle have excellent antireflective properties that affect insect body coloration, reduce surface light reflection, and aid camouflage (*Vukusic and Sambles, 2003*; *Watson et al., 2017*). The first instances of antireflective structures were observed in the

compound eyes of moths, the corneas of which are covered by hexagonally arranged protrusions that are approximately 200 nm apart (*Bernhard, 1965*; *Blagodatski et al., 2015*; *Clapham and Hutley, 1973*). Subsequent research has revealed that antireflective structures are widespread on insect body surfaces, primarily found on compound eyes and wings (*Blagodatski et al., 2015*; *Chotard et al., 2022*; *Ho et al., 2016*; *Chan et al., 2019*; *Stavenga et al., 2006*). These nano- and microstructures on the cuticular layer are believed to possess antireflective properties, effectively minimizing light reflections from their bodies and thus preventing detection by predators (*Blagodatski et al., 2015*; *Chotard et al., 2022*; *Ho et al., 2016*; *Chan et al., 2019*; *Stavenga et al., 2006*).

Leafhoppers (Cicadellidae), one of the largest insect families with over 22,000 species, possess unique extracuticular coating known as brochosome. Brochosomes, typically hollow, honeycomb-like spheres with a diameter ranging from 0.2 to 0.6 μm, are synthesized in the Malpighian tubules, secreted through the hindgut, and applied as a coating on the fresh cuticle following molting (*Rakitov, 2000*; *Rakitov, 2009*). Brochosomes primarily comprise lipids and proteins, with protein content ranging from 45% to 75%, but the exact proteins responsible for brochosome formation remain unknown (*Li et al., 2022*; *Rakitov et al., 2018*; *Yuan et al., 2023*). Some have proposed that brochosomes may serve as a protective layer on the leafhopper's cuticle surface, with a potential role in enhancing the hydrophobicity of their cuticle surface, providing defense against pathogens and predators (*Rakitov and Gorb, 2013a*; *Rakitov, 2009*). Apart from these known function of hydrophobicity and the shedding of brochosomes in helping leafhopper escape spider webs, there is currently no definitive biological evidence supporting other hypothesized functions of brochosomes (*Lin et al., 2021*; *Rakitov and Gorb, 2013a*; *Rakitov and Gorb, 2013b*). The camouflage role of leafhopper brochosomes was first hypothesized by Swain, who proposed that these structures reduce optical reflections to avoid predators (*Swain, 1936*). Subsequent development of bioinspired synthetic analogues has corroborated their antireflective efficacy, with experimental models demonstrating reduced target visibility under simulated predator vision paradigms (*Lei et al., 2020*; *Yang et al., 2017*). Recent mechanistic investigations have further elucidated a geometry-dependent antireflective mechanism intrinsic to brochosome architecture (*Wang et al., 2024*). These findings collectively demonstrate that leafhoppers have evolved a unique antireflection strategy, distinct from conventional insect adaptations, to minimize surface reflectivity. Building upon this foundation, we hypothesize that brochosomes serve as an integumentary camouflage coating, operating through optical signature suppression to enhance crypsis against visual predators and ultimately improve ecological fitness.

In this study, we combined spectrophotometry, electron microscopy, transcriptome analysis, proteome analysis, gene function validation, and leafhopper and jumping spider bioassays to characterize the protein composition and the function of brochosome in leafhopper camouflage. We focused our research on the rice green leafhopper, *Nephotettix cincticeps*, a well-known agricultural pest with an extensive research history (*Hibino, 1996*; *Wei et al., 2018*; *Yan et al., 2021*). Our results revealed the pronounced effectiveness of brochosome coverage in reducing light reflection on leafhoppers *N. cincticeps*, especially within the scope of the UV spectrum. By conducting RNAi experiments, we successfully identified four brochosome structural protein for the first time. The suppression of their synthesis induced changes in brochosome morphology, influencing the optical characteristics of the leafhopper's cuticle surface. Furthermore, our analysis of the evolutionary origin of brochosomes indicates that they likely originated through a process involving duplication–divergence. Our findings support brochosomes as a unique camouflage coating that enables leafhoppers to evade visual predators.

## Results

### Brochosomes are a distinctive coating on the cuticle surface of leafhopper *N. cincticeps*

The cuticle surface of rice green leafhopper *N. cincticeps* is coated with brochosomes (*Figure 1A and B*), which are approximately 350 nm in diameter and featuring a hollow sphere with a honeycomb-like structure (*Figure 1C and D*). The leafhopper *N. cincticeps* possesses two pairs of Malpighian tubules, each consisting of a proximal, distal, and terminal segment (*Figure 1E*). Brochosomes are synthesized in the distal segment of the Malpighian tubule, which exhibits a swollen and rod-shaped appearance (*Figure 1E*). The epithelial cells in this segment contain large spherical nuclei, an extensive

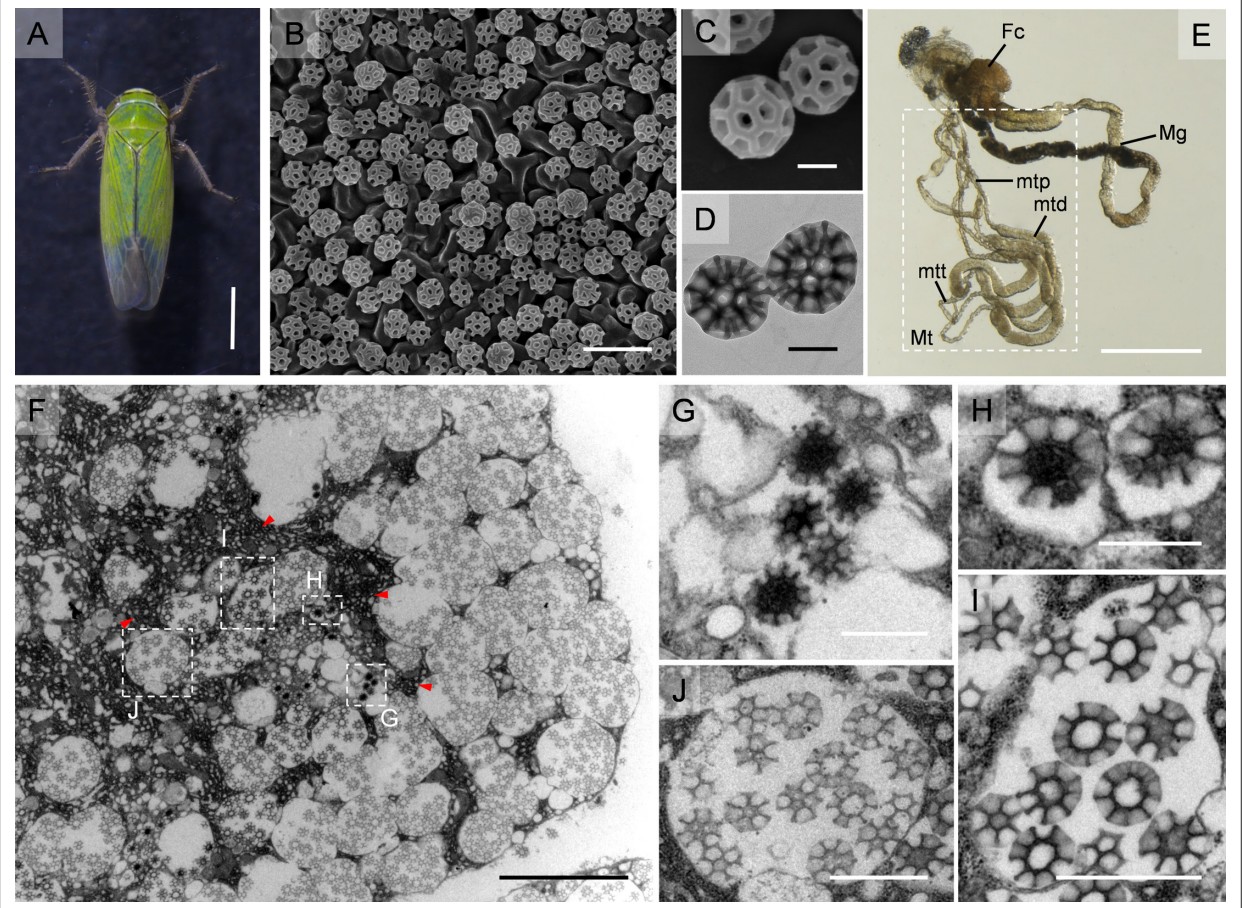

**Figure 1.** Brochosomes are a distinctive coating on the cuticle surface of leafhopper. (**A**) Male adult of the green rice leafhopper *N. cincticeps*. Bar, 1 mm. (**B**) Brochosomes on the surface of the forewing of *N. cincticeps*. Bar, 1 μm. (**C, D**) The morphologies of brochosomes by SEM (**C**) and TEM (**D**). Bar, 200 nm. (**E**) Alimentary tract and the Malpighian tubules of *N. cincticeps*. The leafhopper *N. cincticeps* have two pairs of Malpighian tubules, each divided into proximal segment, distal segment, and terminal segment. The brochosomes are synthesized in the distal segment, which is dilated and rod-shaped. Bar, 1 mm. (**F**) The distal segment epithelial cell displays an extensive rough endoplasmic reticulum and multiple Golgi regions (red arrow) containing developing brochosomes in its basal portion, as well as a number of secretory vacuoles with mature brochosomes near the cell border. Bar, 5 μm. (**G**) The initial stage of the development of brochosome. Bar, 500 mm. (**H**) The two brochosomes that are developing inside the primary vesicles are shown in close-up. Regular invaginations appear on the surface of the growing brochosome at the same time that its matrix separates into a looser core and a denser wall. Bar, 500 nm. (**I**) Larger vesicles containing numerous BS, formed by the fusion of multiple primary vesicles. The surface of the brochosomes has regular cell-like invaginations, and it is surrounded by amorphous flocculent material with a moderate electron density. Bar, 1 μm. (**J**) A vesicle filled with mature brochosomes, each mature brochosome has a spherical inner cavity and a well-swollen outer margin of the septa. Bar, 1 μm. (**E–H**) are the enlargement of boxed area in (**D**). Fc, filter chamber; Mg, midgut; Mt, Malpighian tubules; mtd, distal segment of the Malpighian tubule; mtp, proximal segment of the Malpighian tubule; mtt, terminal segment of the Malpighian tubule. All images are representative of at least three replicates.

rough endoplasmic reticulum, and multiple Golgi regions (*Figure 1E and F*). Within the Golgi-derived vacuole, the brochosomes undergo progressive development, transforming their initially spherical granules into a honeycomb-like surface with closely spaced invaginations (*Figure 1G–I*). Mature brochosomes are stored in secretory vacuoles at the boundaries of the cells and are subsequently secreted into the lumen of the Malpighian tubules (*Figure 1F*).

## The optical properties of the leafhopper's cuticle surface are intricately connected to the brochosome coating

To investigate the interplay between brochosome coverage and the optical features of *N. cincticeps* cuticle surface, we observed the relationship for male and female adults at 5, 10, 15, 20, and 25 days post-eclosion. *N. cincticeps* exhibited a gradual shift in body color with increasing eclosion time,

manifesting clear distinctions between males and females (*Figure 2A*, *Figure 2—figure supplement 1A and B*). Males transformed from light green to dark green, while females progressively transitioned from light green to translucent, displaying an iridescence (*Figure 2A*, *Figure 2—figure supplement 1A and B*). Furthermore, the brightness of the leafhopper cuticle surface under UV light intensified with the extension of post-eclosion time, and the UV reflectance of female cuticle surface was significantly higher than that of males after 15 days post-eclosion (*Figure 2A*, *Figure 2—figure supplement 1A and B*).

To validate the correlation between brochosome coverage, body color, and UV light reflectance, we examined the distribution of brochosomes on the cuticle surface of male and female *N. cincticeps* at various post-eclosion intervals using scanning electron microscopy (SEM). SEM results unveiled a gradual reduction in brochosome coverage on the cuticle surface of both sexes as post-eclosion time progressed. Specifically, male coverage decreased from 90% to approximately 40%, whereas female coverage descended from 90% to around 10% (*Figure 2C*, *Figure 2—figure supplement 1A–C*). Microscopic analyses of the Malpighian tubule revealed that, with prolonged post-eclosion time, the distal segment of the tubule underwent gradual atrophy, particularly pronounced at 15 days post-eclosion. Notably, females exhibited a more marked atrophy in the distal segment compared to males. By 25 days post-eclosion, the distal segment of the Malpighian tubule in females became indistinguishable from other segments (*Figure 2D*, *Figure 2—figure supplement 1A and B*). This denotes a progressive reduction in post-eclosion brochosome synthesis, resulting in a corresponding decrease in brochosome coverage on the cuticle surface. Furthermore, the distribution of brochosomes on the leafhopper cuticle surface may be correlated with its optical characteristics. To substantiate this hypothesis, we analyzed the light reflection characteristics of leafhopper forewings at various post-eclosion time points. The findings revealed a positive correlation between brochosome coverage on the forewings and light reflection values, underscoring the remarkable antireflective attributes of the brochosome coating on leafhopper cuticle surface (*Figure 2E*, *Figure 2—figure supplement 1D and E*). Brochosomal coverage was found to significantly decrease the reflectance of UV light on the leafhopper's surface, reducing it from approximately 30% to 20%. Additionally, the reflectance of visible and infrared light was also notably diminished, dropping from around 20% to 10% (*Figure 2E*, *Figure 2—figure supplement 1D and E*). In summary, the presence of brochosomal coverage resulted in an overall reduction of surface light reflection by approximately 30%, highlighting its substantial antireflective properties.

## Preferential capture of leafhoppers with fewer brochosomes by jumping spiders

Given that UV light serves as a crucial visual cue for arthropod predators, especially common visual hunters like jumping spiders (*Cronin and Bok, 2016*; *Li and Lim, 2005*; *Morehouse et al., 2017*; *Silberglied, 1979*; *Zou et al., 2011*), the function of the brochosome coating on leafhoppers' cuticle surface could be to avoid predation by reduction of UV light reflectance. This prompted us to investigate the effect of brochosome coatings on reducing predation risk in leafhoppers. Predation experiments were conducted using *Plexippus paykulli* (*Figure 2F*, *Figure 2—figure supplement 2*), a common jumping spider in rice fields known to prey on leafhoppers (*Yang et al., 2018*). We observed the spiders' feeding behavior, noting the time of the first attack and the targeted leafhopper, using these metrics to evaluate predation efficiency (*Taylor et al., 2014*; *Walker and Rypstra, 2002*). In the experimental group comprising females and males at 5 days post-eclosion, *P. paykulli* did not exhibit a clear predation preference (*Figure 2G*). However, in the experimental group involving females and males at 25 days post-eclosion, there was a higher probability of predation on female *N. cincticeps* compared to male ones (*Figure 2H*). For male *N. cincticeps* in both the 5-day and 25-day post-eclosion experimental groups, *P. paykulli* showed a stronger inclination towards preying on individuals that were eclosed for 25 days, resulting in significantly shorter predation times compared to those at 5 days post-eclosion (*Figure 2I*). These patterns were consistently observed for female leafhoppers in both the 5 days and 25 days post-eclosion (*Figure 2J*). The cumulative findings strongly suggest a distinct preference of the jumping spider *P. paykulli* for individuals with lower brochosome coverage on the cuticle surface. Based on these experiments, we proposed the hypothesis that the brochosome coating on the leafhopper cuticle surface may mitigate predation risk by reducing surface light reflection, particularly in the UV spectrum.

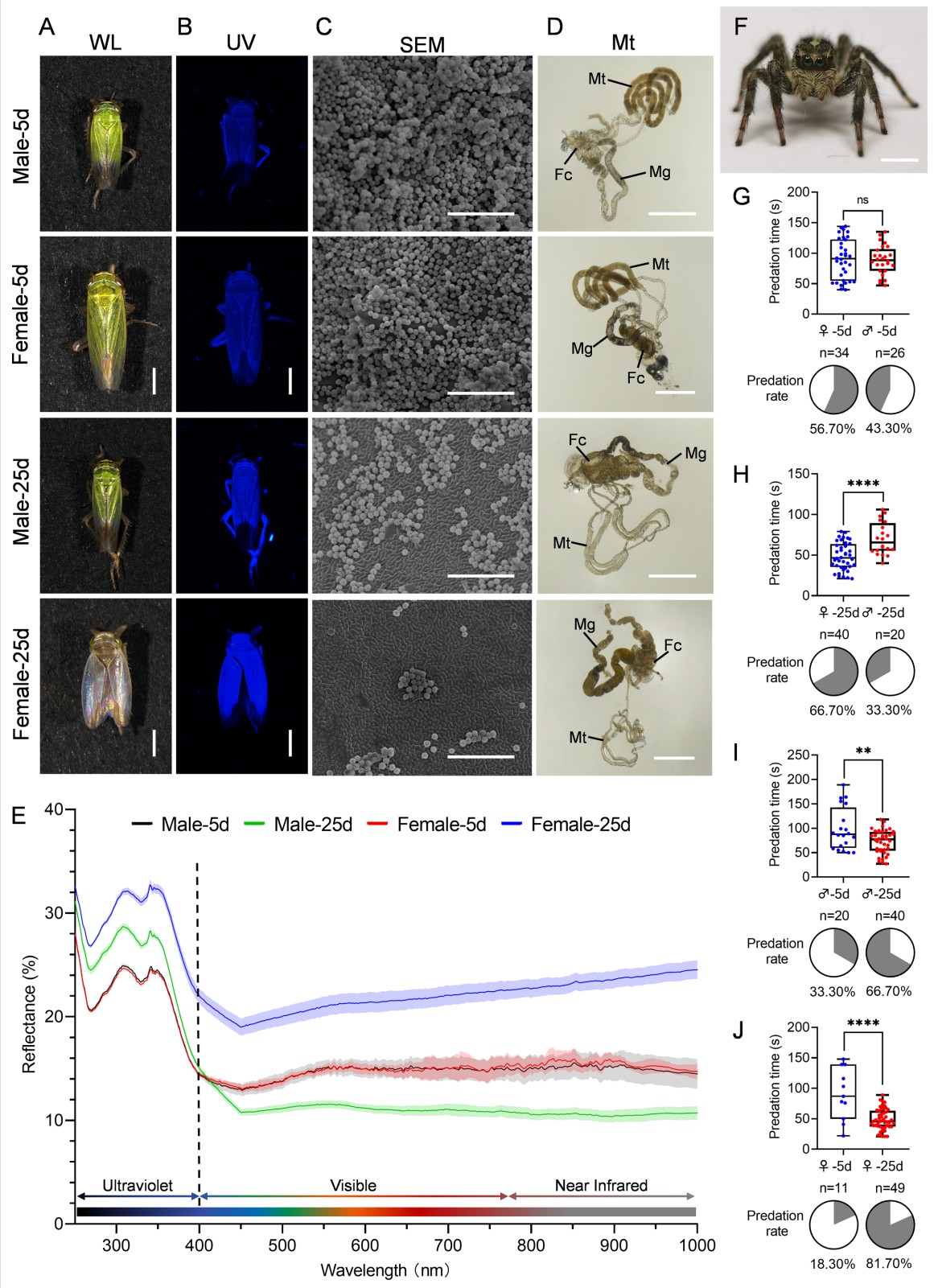

**Figure 2.** The distribution of brochosomes on the cuticle surface of leafhopper *N. cincticeps* are associated with predation by jumping spiders. (**A,** **B**) Images of leafhopper *N. cincticeps* males and females in white (**A**) and ultraviolet light (**B**) at 5 and 25 days post-eclosion, respectively. Bar, 1 mm. (**C,** **D**) The distribution of brochosomes on the surface of a forewing (**C**) and morphological changes of the Malpighian tubules (**D**) of *N. cincticeps* males and females at 5 and 25 days post-eclosion. Bar, 5 μm in (**C**); Bar, 1 mm in (**D**). (**E**) Reflectance spectra of female and male forewing of *N. cincticeps* at 5

*Figure 2 continued on next page*

*Figure 2 continued*

and 25 days post-eclosion. The average curve is based on data from five independent samples, with standard deviation represented by the shaded area. (**F**) Images of the jumping spider *P. paykulli*. Bar, 2 mm. (**G–J**) Jumping spiders prefer leafhoppers with little brochosome covering as food. In predation experiment, jumping spiders offered *N. cincticeps* male and female at 5 days post-eclosion (**G**), male and female at 25 days post-eclosion (**H**), males at 5 and 25 days post-eclosion (**I**), and females at 5 and 25 days post-eclosion (**J**). Data on predation times are displayed using the traditional box and whisker shapes. All box plots with whiskers represent the data distribution based on five number summary statistics (maximum, third quartile, median, first quartile, minimum), each dot in box plot represents an independent experiment. **$p<0.01$, ****$p<0.0001$, ns no significance, Statistical significance was determined by unpaired *t*-test with Welch's correction method. Predation preference is shown in the pie chart. All images are representative of at least three replicates.

The online version of this article includes the following figure supplement(s) for figure 2:

**Figure supplement 1.** Influence of brochosome coating on optical properties of *N. cincticeps* cuticle surface.

**Figure supplement 2.** Frontal view (**A**), top view (**B**), and side view (**C**) of jumping spider *P. paykulli*.

## Identifying the major structural proteins of brochosomes

Despite the initial discovery of brochosomes in the early 1950s and the confirmation in the 1960s that they are protein-lipid particles (*Gouranton and Maillet, 1967*), the specific protein composition of brochosomes remains unknown. Recent studies suggested that they are primarily composed of brochosomins (BSM) and brochosome-associated proteins (BSAP) (*Li et al., 2022*; *Rakitov et al., 2018*). BSM, a novel class of secreted proteins with molecular weights ranging from 21 to 40 kDa, is considered the major structural component of brochosomes (*Rakitov et al., 2018*). Proteomics studies have identified the proteins composing brochosomes, but the specific proteins involved in their formation remain unclear.

To further elucidate the essential protein components of brochosomes, transcriptomic and proteomic data derived from brochosome of *N. cincticeps* were comprehensively analyzed. Based on our prior integrated analysis of the brochosome transcriptome and proteome (*Wu et al., 2023*), in conjunction with literature search results, we selected 50 candidate genes for functional analysis using RNAi-mediated gene silencing. Through RNAi experiments, we successfully identified four genes encoding brochosome structural proteins that led to morphological changes in brochosomes (*Figure 3A*). According to the coding sequence (CDS) length of these genes, the four proteins were named as BSM-1 to BSM-4 (GenBank accession numbers PP273097, PP273098, PP273099, PP273100) (*Figure 3B*). Homology analysis suggested that BSM-coding genes might be paralogous (*Figure 3—figure supplement 1A*). Although BSM-2 and BSM-3 exhibited low sequence homology, their protein structures were highly conserved (*Figure 3—figure supplement 1B and C*). To validate the functions of these four BSM-coding genes, we synthesized dsRNA from two nonoverlapping regions of each gene to test for off-target effects (*Figure 3—figure supplement 2A*). RT-qPCR revealed that both individual and mixed dsRNA injections effectively suppressed the expression of the BSM-coding genes, with mixed injections achieving higher efficiency (*Figure 3C–F*, *Figure 3—figure supplement 2C–J*). SEM observations showed that nonoverlapping fragments of dsRNA targeting the same gene induced similar morphological changes in brochosomes. These were characterized by increased diameters and deformed honeycomb-like structures (*Figure 3A*, *Figure 3—figure supplement 2B*). Statistical analysis of SEM data from mixed dsRNA injections indicated a 60–70% reduction in brochosome distribution area and a 20% incidence of morphologically abnormal brochosomes compared to dsGFP controls (*Figure 3—figure supplement 3*). Additionally, temporal and spatial expression analyses demonstrated that the BSM-coding genes were specifically expressed in the Malpighian tubules (*Figure 3G–J*) and exhibited relatively stable expression during the early post-eclosion period, followed by a gradual decline after 10 days (*Figure 3K–N*). By 25 days post-eclosion, the expression of BSM-coding genes declined to around 10% in females and 30–40% in males, respectively (*Figure 3K–N*). This phenomenon is consistent with our initial microscopic observations, suggesting that the gradual reduction in brochosome synthesis contributes to the decrease of brochosome coverage on the cuticle surface of *N. cincticeps* after adulthood.

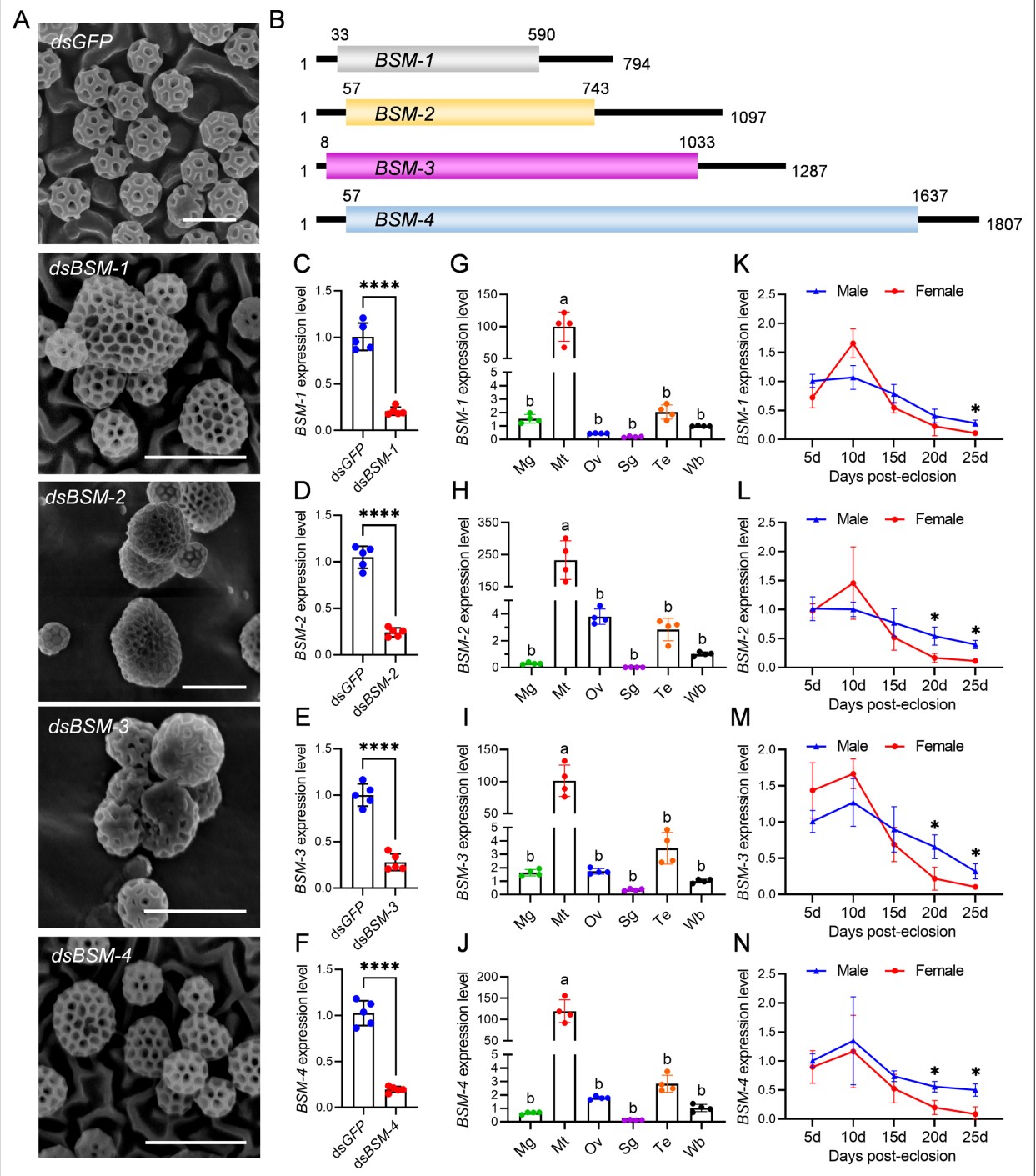

**Figure 3.** Identification of brochosome structural proteins. (**A**) Morphology of brochosomes on the forewing of leafhopper *N. cincticeps* at 7 days post-microinjection with dsRNA mix targeting two nonoverlapping regions of each BSM gene. Bar, 500 nm. (**B**) Gene structures of *BSM-1*, *BSM-2*, *BSM-3,* and *BSM-4*. (**C–F**) Transcription levels of BSM-1 (**C**), BSM-2 (**D**), BSM-3 (**E**), and BSM-4 (**F**) at 7 days post-microinjection with dsRNA mix targeting two nonoverlapping regions of each BSM gene. Each data point represents the result of one independent experiment. Results were obtained from 5 independent experiments. Each data point represents the result of one independent experiment. (**G–J**) The abundance of BSM transcripts in different tissues and whole bodies of *N. cincticeps* was determined by RT-qPCR. Notably, *BSM-1* (**G**), *BSM-2* (**H**), *BSM-3* (**I**), and *BSM-4* (**J**) exhibited specific expression in the Malpighian tubules. Results were obtained from 4 independent experiments. Each data point represents the result of one independent experiment. (**K–N**) The expression patterns of *BSM-1* (**K**), *BSM-2* (**L**), *BSM-3* (**M**), and *BSM-4* (**N**) transcripts were examined in male and female leafhopper at 5, 10, 15, 20, and 25 days post-eclosion. Results were obtained from 3 independent experiments. For (**C–N**), data shown are

*Figure 3 continued on next page*

*Figure 3 continued*

mean ± SD values. *p<0.05; **p<0.01; ***p<0.001; ****p<0.0001; ns no significance (**C–F**, two-tailed Student's *t*-test; **G–J**, one-way ANOVA; **K–N**, two-way ANOVA). All images are representative of at least three replicates.

The online version of this article includes the following figure supplement(s) for figure 3:

**Figure supplement 1.** Homology analysis of BSM-encoding genes.

**Figure supplement 2.** BSM1-4 are brochosomal structural proteins.

**Figure supplement 3.** Inhibition of BSM gene expression reduces brochosome distribution on *N. cincticeps* cuticle surface (**A**) and alters their morphology (**B**). Results were obtained from 15 independent experiments. Each data point represents the result of one independent experiment. The presented data are expressed as mean ± SD values.

## Brochosome coating diminishes light reflection and facilitates predator avoidance

To further investigate the function of brochosomes in leafhopper cuticle surface associated with light reflection and predator avoidance, we implemented RNAi to simultaneously suppress the expression of all four BSM-coding genes with ds*BSM* mixture injection. RT-qPCR results demonstrated a substantial downregulation in the expression of these genes compared to the ds*GFP* control (*Figure 4F*). Notably, under UV light, both male and female leafhoppers exhibited a significant increase in UV reflection on their cuticle surface following ds*BSM* treatment (*Figure 4A*). Spectral data corroborated these findings, showing elevated emission values in the UV spectrum on the leafhopper's cuticle surface (*Figure 4G*). SEM showed that dsBSM treatment significantly altered brochosome morphology and distribution (*Figure 4B and C*, *Figure 4—figure supplement 1A and B*). The distribution area decreased by 80%, and deformed brochosomes accumulated, contrasting with the uniform distribution in the dsGFP treatment (*Figure 3—figure supplement 3A*). Additionally, nearly 30% of brochosomes exhibited significant morphological changes following dsBSM treatment (*Figure 3—figure supplement 3B*). Transmission electron microscopy of the Malpighian tubules indicated numerous Golgi-derived vacuoles without brochosome distribution in epithelial cells after ds*BSM* treatment (*Figure 4D and E*, *Figure 4—figure supplement 1C and D*). Predation experiments with jumping spiders revealed an increased predation preference for leafhoppers treated with ds*BSM*. For male leafhoppers, the proportion attacked of the predator was 70%, with a significantly shorter predation time (71.3 s) compared to the ds*GFP* control (100.4 s) (*Figure 4H*). Similarly, for females, there was a higher predation rate (68.3%) and a significantly shorter predation time (79.6 s) for ds*BSM*-treated groups compared to the ds*GFP* ones (30% and 109.9 s, respectively) (*Figure 4I*).

## Normal structure of the brochosome correlates with its antireflective properties

To further elucidate the correlation between brochosome morphology and its optical performance, we collected brochosomes from the wings of leafhoppers treated with ds*GFP* and ds*BSM*, respectively. Subsequently, these brochosomes were applied to the wings of brown planthoppers and quartz slides. SEM observations revealed that the coverage of brochosomes derived from both ds*GFP* and ds*BSM* treatments on the wings of brown planthoppers and quartz slides was nearly identical (*Figure 4—figure supplement 2*). Spectral measurements indicated that the application of brochosomes from both ds*GFP* and ds*BSM* treatments effectively reduced the reflectance of the brown planthopper wings and quartz slides. It should be noted that brochosomes from the ds*GFP* treatment exhibited significantly higher antireflective performance compared to those from the ds*BSM* treatment (*Figure 4J and K*). Additionally, purified BSM proteins applied to quartz glass did not show improved antireflective performance over purified GST protein (*Figure 4—figure supplement 3*), indicating a strong correlation between brochosome geometry and optical performance.

## Brochosomes as a camouflage coating for the leafhoppers in the family Cicadellidae

Based on the aforementioned findings, brochosomes of *N. cincticeps* can be considered as an antireflective stealth coating against visual recognition by the predator *P. paykulli*. However, it remains unknown whether brochosomes also serve as a stealth coating in other leafhopper species. Therefore,

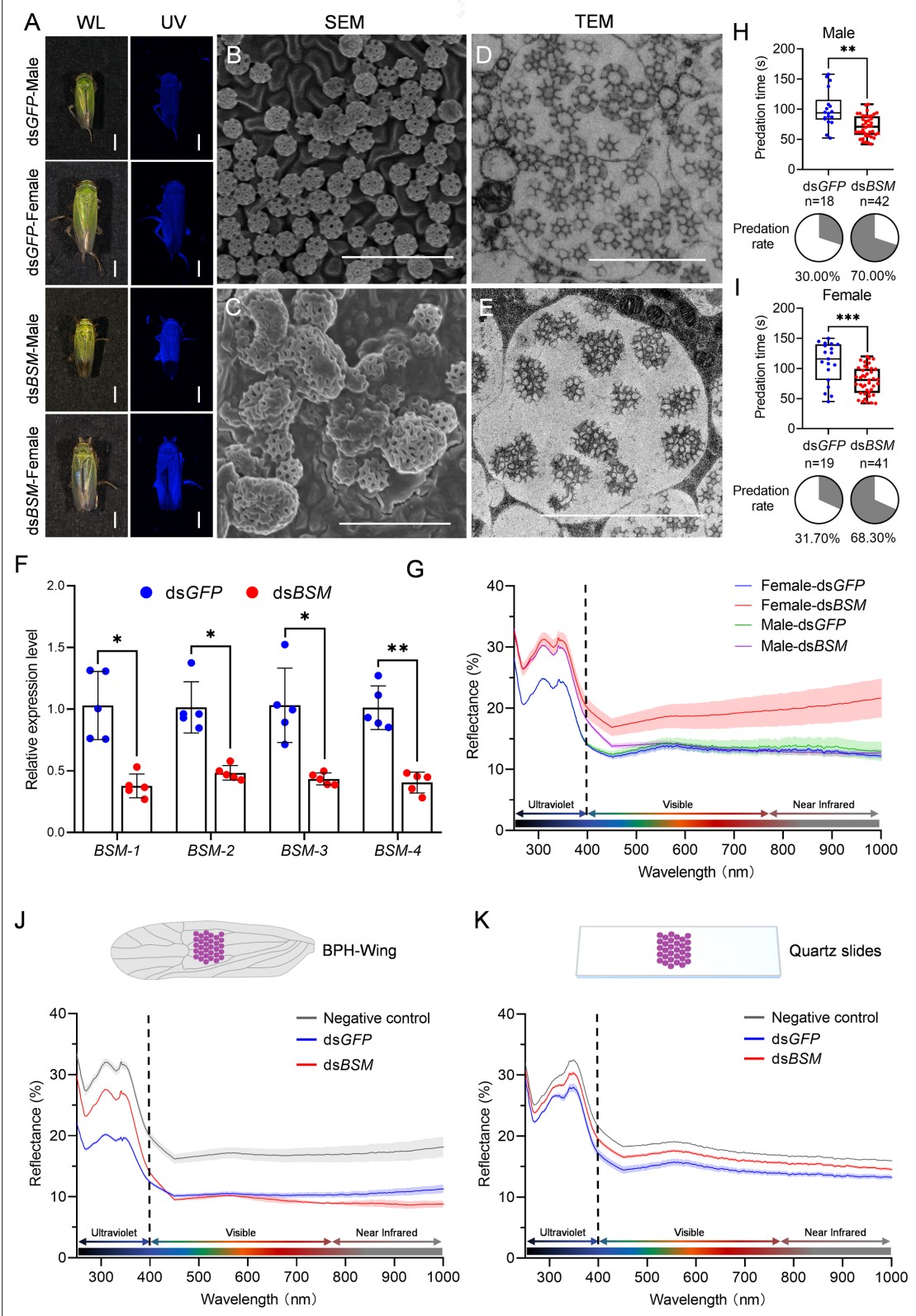

**Figure 4.** RNAi inhibits brochosome synthesis, alters brochosome morphology, and influences predation in jumping spiders. (**A**) Images of leafhopper *N. cincticeps* males and females in white and ultraviolet light after ds*GFP* or ds*BSM* treatment, respectively. Bar, 1 mm. (**B, C**) Morphology of the brochosome on the forewings of leafhoppers after ds*GFP* (**B**) and ds*BSM* (**C**) treatment. Bar, 2 μm. (**D, E**) Morphology of the brochosome in the distal segment epithelial cells of Malpighian tubules after ds*GFP* (**D**) and ds*BSM* (**E**) treatment. Bar, 2 μm. (**F**) The transcript levels of *BSM-1*, *BSM-2*, *BSM-3*,

*Figure 4 continued on next page*

*Figure 4 continued*

and *BSM-4* at 7 days after ds*GFP* and ds*BSM* treatment. Results were obtained from 5 independent experiments. Each data point represents the result of one independent experiment. The presented data are expressed as mean ± SD values. Statistical significance is denoted as *$p<0.05$ and **$p<0.01$, determined by two-way ANOVA. (**G**) Reflectance spectra of female and male forewing of *N. cincticeps* at 7 days after ds*GFP* and ds*BSM* treatment. The average curve is based on data from five independent samples, with standard deviation represented by the shaded area. (**H, I**) Jumping spiders prefer to prey on ds*BSM*-treated leafhoppers. Predation efficiency and preference of jumping spiders on males (**H**) and females (**I**) *N. cincticeps* after ds*GFP* and ds*BSM* treatment in the predation experiment. Data on predation times are displayed using the traditional box and whisker shapes. All box plots with whiskers represent the data distribution based on five number summary statistics (maximum, third quartile, median, first quartile, minimum), each dot in box plot represents an independent experiment. **$p<0.01$, ***$p<0.001$, Statistical significance was determined by unpaired *t*-test with Welch's correction method. Predation preference is shown in the pie chart. (**J, K**) The morphology of brochosomes is related to their optical performance. Collect brochosomes treated with ds*GFP* and ds*BSM* separately, apply them to brown planthopper wings (**J**) and quartz slides (**K**), and set up quartz slides or brown planthopper wings treated solely with acetone as negative controls. The average curve is based on data from five independent samples, with standard deviation represented by the shaded area. All images are representative of at least three replicates.

The online version of this article includes the following figure supplement(s) for figure 4:

**Figure supplement 1.** Differences in the synthesis, morphology, and distribution of brochosomes following RNAi treatment.

**Figure supplement 2.** Correlation between brochosome morphology and optical properties.

**Figure supplement 3.** The purified BSM protein lacks antireflective properties.

we extended our investigation to include additional leafhopper species in the family Cicadellidae, including *Recilia dorsalis*, *Empoasca onukii*, and *Psammotettix alienus*. Brochosomes were carefully removed from the forewings of leafhopper specimens using acetone. SEM observations confirmed the effective removal of brochosome coatings from *N. cincticeps* forewings (*Figure 5—figure supplement 1*). Subsequent quantitative analysis demonstrated a statistically significant enhancement in UV reflectance on acetone-treated *N. cincticeps* forewings compared to untreated controls (*Figure 5A and B*). In addition to *N. cincticeps*, we observed a considerably higher brightness of acetone-treated forewings compared to the untreated ones under UV light for *R. dorsalis*, *E. onukii*, and *P. alienus* (*Figure 5A*). This result suggests that brochosomes as an antireflective stealth coating may be commonly existed in various species of the family Cicadellidae.

Previous research indicated that the brochosome is a unique secretion in Cicadellidae (*Rakitov and Gorb, 2013b*; *Rakitov et al., 2018*). To investigate the conservation of genes linked to brochosome synthesis in Cicadellidae, we systematically screened these four BSM coding genes across 116 other Hemipteran insect species (*Supplementary file 2*). Results of homology analysis reveal that BSM coding genes are orphan genes restricted to the clade of Membracoidea. However, there are variations in the species distribution of different BSM coding genes. In Membracoidea species, homologous genes for *BSM-3* and *BSM-4* are observed, while *BSM-2* is identified in Cicadellidae insects, and *BSM-1* in Deltocephalinae insects (*Figure 5C*). An intriguing observation is that in the majority of Cicadellidae species with confirmed brochosome distribution, homologous gene distribution for *BSM-2* is detectable. In Cicadellidae species without brochosome distribution and in the majority of Membracidae, *BSM-2* genes are largely absent (*Figure 5C*). Since Membracidae evolved from Cicadellidae, we hypothesize that BSM homologous gene duplication/loss may be a key factor in brochosome formation/loss. In conclusion, our data suggest that in Cicadellidae, brochosome is synthesized by a conserved group of BSM coding genes, making it a widely distributed antireflective camouflage coating in Cicadellidae.

## Discussion

Leafhoppers exhibit a distinctive grooming behavior, resulting in the deposition of a unique extracuticular coating known as brochosome (*Rakitov, 2009*). However, the precise characteristics and exact adaptive significance of this specialized coating remain largely unknown. Although they are widely believed to confer hydrophobicity, protecting leafhoppers from water and their own excreta (*Rakitov and Gorb, 2013a*), similar functionalities are attributed to cuticular waxes on the external surfaces of various insects, including leafhoppers (*Andersen, 1979*; *Bello et al., 2022*). Consequently, the functionalities of brochosomes may extend beyond this conventional understanding. In this study, we demonstrated that brochosomes exhibit robust antireflective properties, decreasing the reflectance of the leafhopper cuticle surface, particularly in the UV spectral range. This reduction may

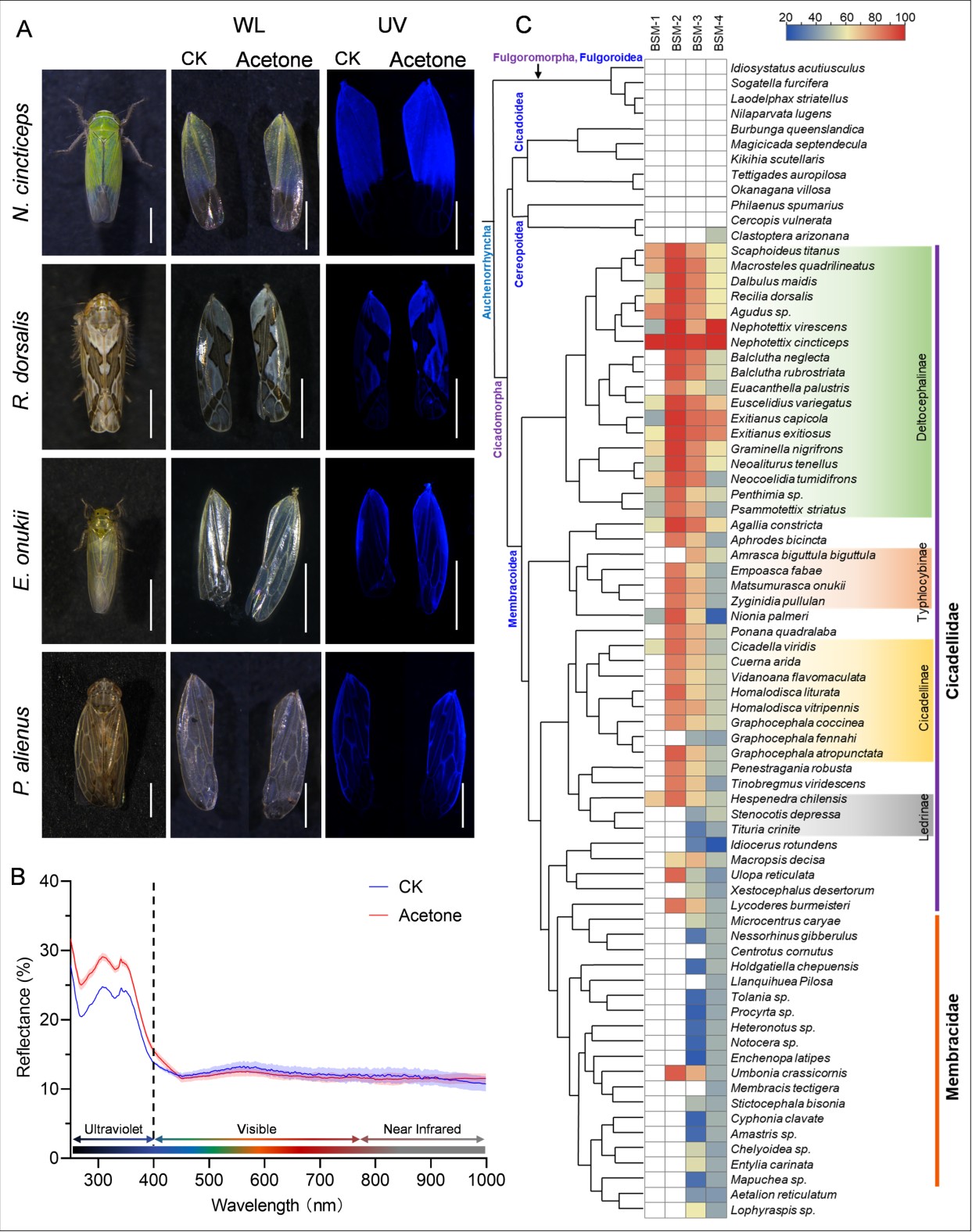

**Figure 5.** Brochosome coating on Cicadellidae cuticle surface is essential for their optical qualities. (**A**) Images of forewings of four leafhopper species before and after acetone treatment in white and ultraviolet light. Bar, 1 mm. (**B**) Reflection spectra of the forewings of the *N. cincticeps* before and after acetone treatment. (**C**) Phylogenetic associations between the BSM and phylogeny of Hemipteran lineages. The left column describes the phylogeny of 76 representative species of Hemiptera with BSM proteins. In the right column, the presence of BSM-encoded genes in different Hemiptera species is

*Figure 5 continued on next page*

*Figure 5 continued*
illustrated along with their homology analysis with *N. cincticeps*. White indicates the absence of related genes; the color gradient represents the degree of nucleotide sequence similarity. All images are representative of at least three replicates.

The online version of this article includes the following figure supplement(s) for figure 5:

**Figure supplement 1.** SEM analysis of *N. cincticeps* forewing before and after acetone treatment.

reduce visibility to visual predators, contributing to the evasion and predation avoidance (*Vukusic and Sambles, 2003*; *Watson et al., 2017*). Thus, brochosomes function as a natural camouflage coating on the leafhopper cuticle surface, playing a pivotal role in reducing leafhoppers' visibility to visual predators and offering a distinct advantage for their survival in predation scenarios as illustrated in *Figure 6*.

The present work has elucidated the function of brochosome as a natural antireflective camouflage coating on leafhoppers' extracuticular, with its coverage directly influencing the antireflective properties of the cuticle surface (*Figures 2A–E and 4A–G*). Previous studies suggested that arthropods employ various nanostructures formed by proteins, lipids, wax, or adhesive substances on their cuticle surfaces to achieve antireflectivity (*Silberglied, 1979*). These nanostructures, initially identified in the compound eyes of moths, have been subsequently found to be widely distributed on the cuticle surfaces of insects (*Bernhard, 1965*; *Blagodatski et al., 2015*; *Chotard et al., 2022*; *Clapham and Hutley, 1973*; *Ho et al., 2016*; *Chan et al., 2019*; *Stavenga et al., 2006*). The nano- and microstructures on insect cuticle surfaces generate a gradual change in refractive index near the surface, leading to antireflectivity (*Raut et al., 2011*). The degree of reflection reduction depends on the shape of these nanostructures, studies in *Drosophila melanogaster* have confirmed that modifying the morphology of nanostructures in compound eyes directly impacts their optical performance (*Kryuchkov et al., 2020*). In this study, a thorough examination of the structural protein composition of brochosome, coupled with RNAi-mediated suppression of BSM coding gene expression, yielded brochosomes with regular structure (ds*GFP* treatment) and structurally abnormal brochosomes (ds*BSM* treatment) (*Figure 4B–E*). Subsequent in vitro experiments provided further evidence of the correlation between the normal structure of brochosome and its antireflective performance (*Figure 4J*

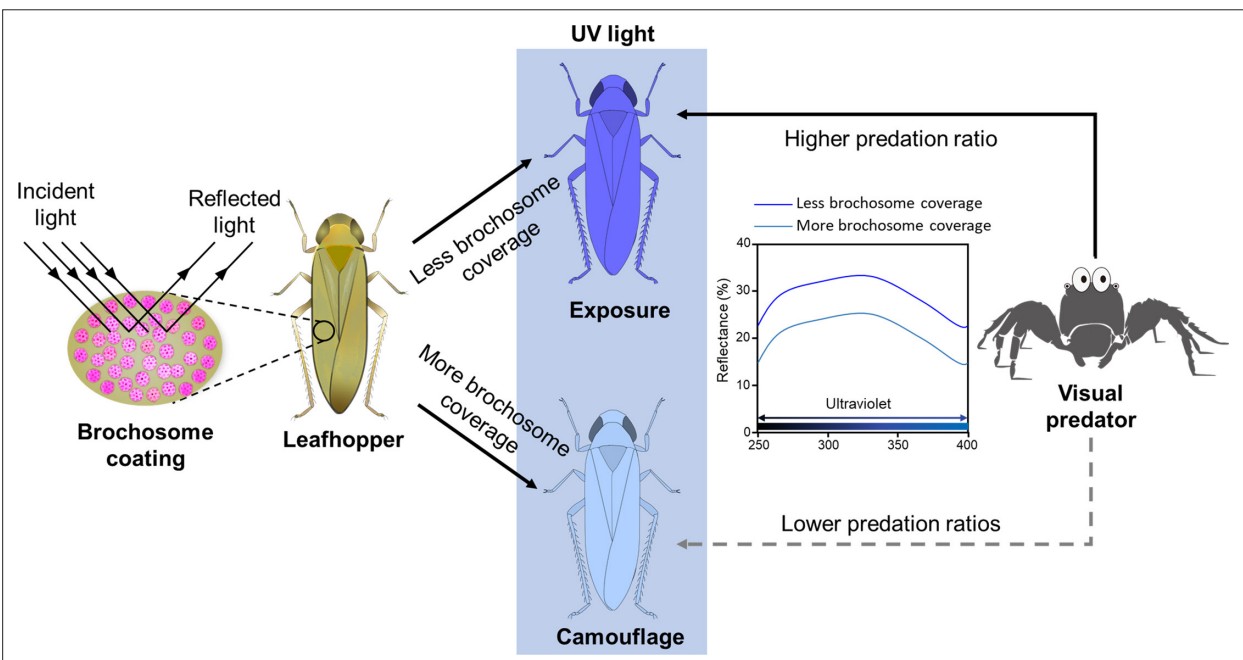

**Figure 6.** Brochosomes serve as an antireflective camouflage coating on the cuticle surface of leafhoppers. Brochosomes effectively reduce the reflection of various wavelengths of light, particularly in the UV region. UV light is a crucial visual cue for numerous visual predators to identify and locate prey. Brochosomes efficiently decrease the UV light reflection on the surface of leafhoppers, thereby reducing their exposure risk to visual predators and facilitating evasion of visual predation.

and K). This correlation is likely attributed to the fact that the diameter of the hollow pits in brochosome structure (approximately 100 nm) is considerably smaller than most spectral wavelengths (*Burks et al., 2023*; *Rakitov, 2009*). As light passes through these small pits, it undergoes diffraction, while light passing through the ridges of brochosome induces scattering. The interference of diffracted and scattered light between different pits and ridges contributes to the observed extinction features in brochosome (*Raut et al., 2011*). Biomimetic material and mathematical modeling analyses have clarified the relationship between the extinction features of brochosome and the spacing between hollow pits (*Banerjee et al., 2023*; *Lei et al., 2020*; *Wang et al., 2024*; *Yang et al., 2017*). Therefore, the regular morphology of brochosome is a robust guarantee for its excellent antireflective performance.

Moreover, we noticed a distinct contrast in the antireflective capabilities when applying brochosomes with regular and irregular structures onto different substrates (quartz glass and brown planthopper wings), especially in the UV region (*Figure 4J and K*). This indicates that the reduction in reflected light is not only related to the structure of brochosome but also to other factors. SEM observations revealed that some brochosome is embedded in the grooves of the leafhopper cuticle surface or brown planthopper wings, which is not observed on the smooth surface of quartz glass (*Figure 4B*, *Figure 4—figure supplement 2*). When the wavelength of light is higher than the size of the structure and the surface has a gradient refractive index, light interacts entirely with the rough surface, causing the light to gradually bend (*Raut et al., 2011*). Combined with the complex structure formed by brochosomes and nanostructures on the cuticle surface, this result in multiple internal reflections of light within these structures, increasing light absorption and significantly enhancing antireflective performance. Furthermore, the aromatic ring structures, conjugated double bonds, and peptide bonds in the proteins of both the cuticle and brochosomes may contribute to the absorption of UV light, providing another potential reason for their antireflective capabilities in the UV spectral region. Therefore, the antireflective performance of brochosome on the insect cuticle surface might be a result of the combined effect of brochosome and cuticular nanostructures.

UV light serves as a crucial visual cue for various insect predators, enhancing foraging, navigation, mating behavior, and prey identification (*Cronin and Bok, 2016*; *Morehouse et al., 2017*; *Silberglied, 1979*). Predators such as birds, reptiles, and predatory arthropods often rely on UV vision to detect prey (*Church et al., 1998*; *Li and Lim, 2005*; *Zou et al., 2011*). However, UV reflectance from insect cuticles can disrupt camouflage, increasing the risk of detection and predation, as natural backgrounds like leaves, bark, and soil typically reflect minimal UV light (*Endler, 1993*; *Li and Lim, 2005*; *Tovee, 1995*). To mitigate this risk, insects often possess antireflective cuticular structures that reduce UV and broad-spectrum light reflectance. This strategy is widespread among insects, including cicadas, dragonflies, and butterflies, and has been shown to decrease predator detection rates (*Hooper et al., 2006*; *Siddique et al., 2015*; *Zhang et al., 2006*). For example, the compound eyes of moths feature hexagonal protuberances that reduce UV reflectance, aiding nocturnal concealment (*Blagodatski et al., 2015*; *Stavenga et al., 2006*). In butterflies, UV reflectance from eyespots on wings can attract predators, but reducing UV reflectance or eyespot size can lower predation risk and enhance camouflage (*Chen et al., 2023*; *Lyytinen et al., 2004*). Hence, the reflection of UV light from the insect cuticle surface increases the risk of predation by disrupting camouflage (*Tovee, 1995*). In this study, we utilized the jumping spider *P. paykulli* to explore the impact of the brochosome coating on the leafhopper's camouflage against predation. Hunting spiders, such as jumping spiders and wolf spiders, are primary predators of leafhoppers (*Liu et al., 2015*; *Oraze and Grigarick, 1989*), and the wavelength of UV light plays a crucial role as a visual cue for jumping spiders in identifying and locating their prey compared to the other wavelengths (*Li and Lim, 2005*; *Zou et al., 2011*). Our results revealed that *P. paykulli* displayed a preference for preying on *N. cincticeps* with a lower brochosome coverage on their cuticle surface (*Figures 2G–J and 4H–I*). Thus, we hypothesize that the combination of brochosome and leafhopper cuticular microstructures exhibits superior antireflective performance in the UV region, effectively reducing UV reflection from leafhopper bodies and consequently diminishing the predator's ability to identify them. Additionally, spectral measurement data indicated that the wing reflection spectrum of the brown planthopper (*Figure 4J*) closely resembled the body spectrum of 25-day-old female *N. cincticeps* (*Figure 2E*). It is possible that, in the same living environment, other Hemipteran insects would be more conspicuous to visual predators than leafhoppers. Brochosome coating could provide an advantage for leafhoppers to evade visual predators, thereby enhancing their survival rates in interspecific competition. This adaptation could potentially

favor the survival of Cicadellidae in environments with high predation pressure, allowing them to thrive in a wider range of environments. Additionally, within the leafhopper population, the brochosome coverage on the cuticle surface of older individuals was notably lower than that of younger individuals, making elderly leafhoppers more susceptible to be detected and captured by predators (*Figure 2F–I*, *Figure 2—figure supplement 1*). This may help eliminate older individuals who have lost their reproductive capacity in the population, thereby ensuring the vitality and reproductive power of the population. Therefore, we consider brochosome, as a form of antireflective camouflage coating on the leafhopper's cuticle surface, represents a more advanced evolutionary strategy against visual predation.

We successfully identified four proteins essential for the structure of brochosomes for the first time (*Figure 3*). These BSM-encoded genes exhibit typical orphan gene characteristics, consistent with previous findings (*Li et al., 2022*; *Rakitov et al., 2018*; *Yuan et al., 2023*). Orphan genes originate from processes like de novo evolution, horizontal gene transfer (HGT), and duplication–divergence (*Light et al., 2014*; *Tautz and Domazet-Lošo, 2011*). To elucidate the evolutionary origin of the four identified BSM genes, we aligned them with all prokaryotic genomes in NCBI and approximately 1000 arthropod genomes. Interestingly, the homologous genes of these BSM genes were exclusively found in leafhopper genomes in the family Cicadellidae, suggesting that the origin of these BSM genes might not be due to de novo evolution or horizontal gene transfer. Combining the species distribution of the four identified BSM genes (as well as their homology) (*Figure 5C*), we hypothesize that the BSMs were potentially originated from a process involving duplication–divergence. In this scenario, a new gene would be created through gene duplication or transposition, undergoing fast adaptive evolution and losing similarity to the original gene (*Light et al., 2014*; *Tautz and Domazet-Lošo, 2011*). For instance, BSM-4 is widely present in Membracoidea species and also has homologous genes in *Clastoptera arizonana* of Cereopoidea, indicating that BSM-4 may have a common ancestral gene in Cicadomorpha. The specialization of the distal segment of the Malpighian tubule is a characteristic shared by the three major lineages of the infraorder Cicadomorpha: Cercopoidea (spittle-bugs), Cicadoidea (cicadas), and Membracoidea (leafhoppers and treehoppers) (*Rakitov, 2002*). Cicadas coat the surface of their burrows with secretions from the Malpighian tubules, spittlebug nymphs have their integuments covered with foam synthesized by the Malpighian tubules, and leafhoppers coat their integuments with brochosomes synthesized by the Malpighian tubules (*Chang et al., 2019*; *Rakitov, 2002*). The different demands for secretions from the Malpighian tubules among these insects may be the main reason for the rapid adaptive evolution of BSM genes.

Among Auchenorrhyncha insects, most Cicadellidae (leafhoppers) secrete brochosomes, while many Membracidae (treehoppers) exhibit grooming behavior but lack specialized setae (*Rakitov, 1996*). Considering that Membracidae are derived from Cicadellidae (*Dietrich et al., 2001*), the formation of the symbiotic relationship between treehoppers and ants might due to the loss of brochosome synthesis capability. Ants create a protective environment around treehoppers, deterring threats and reducing predation pressure, providing a relatively secure living environment, leading to lower predation pressure compared to leafhoppers (*Del Claro and Oliveira, 2000*; *Nelson and Mooney, 2022*). Consequently, the need for the protective disguise represented by brochosomes is likely diminished. We hypothesize that environmental changes drive brochosome evolution, supported by the distribution of BSM-2 and BSM-3 paralogs in Auchenorrhyncha. BSM-3 homologs are widespread in Auchenorrhyncha, while BSM-2 is restricted to brochosome-secreting Cicadellidae and absent in most Membracidae. The high structural homology between BSM-2 and BSM-3, despite low sequence similarity, suggests that BSM-2 arose from BSM-3 duplication. We propose that gene duplication/loss events are the primary factors underlying differences in brochosome synthesis between Cicadellidae and Membracidae. Gene duplication/loss is a common mechanism for functional gain/loss in eukaryotes, promoting protein diversity and redundancy (*Wong and Belov, 2012*). Thus, gene duplication may be a key driver of brochosome formation and functional diversity in leafhoppers.

In summary, our research has underscored the indispensable role of brochosomes as an essential antireflective camouflage coating for leafhoppers, conferring a significant advantage in evading visually oriented predators. We have identified four key BSM encoding genes, a groundbreaking discovery in brochosome research, which suggests that gene duplication may be involved in the formation of these adaptive structures. Our study not only deepens our understanding of leafhoppers' antipredator strategies but also sheds light on the distinct evolutionary trajectories employed

by these insects within their ecological niche. This contributes to the development of biomimicry and the advancement of state-of-the-art camouflage technologies. Furthermore, while our study has provided valuable insights into the structure and function of brochosomes, there are still gaps in our knowledge regarding the evolutionary origins of BSMs. Future research should aim to fill these gaps as a more comprehensive understanding of BSMs and their role in brochosome formation will enhance our appreciation of the complex evolutionary processes that have shaped these unique structures. Additionally, the analysis of BSMs in a broader range of leafhopper species will help elucidate the relationship between BSM composition and the morphological and functional diversity of brochosomes, further informing our understanding of the ecological and evolutionary dynamics of this insect group.

## Materials and methods

### Insect

The leafhopper *N. cincticeps* and *R. dorsalis* adults were collected from a rice field in Jiaxing, Zhejiang Province, China, in September 2020. *P. alienus* adults were collected from a wheat field in Shenyang, Liaoning Province, China, in May 2020. *E. onukii* adult were collected from a tea garden in Fuzhou, Fijian Province, China, in May 2020. The identity of the leafhopper species was confirmed using stereomicroscopy and mitochondrial cytochrome oxidase subunit 1 (CO1) sequence analysis (*Supplementary file 1*). Collected leafhopper *N. cincticeps* and *R. dorsalis* are maintained in insect-proof greenhouses at 26 ± 1°C under a 16:8 h light:dark photoperiod and 50 ± 5% relative humidity on rice variety TaiChung Native 1 (TN1).

The jumping spider *P. paykulli* were collected from a rice field in Ningbo, Zhejiang Province, China. Species identification was confirmed by assessing its morphological characteristics using stereomicroscopy and CO1 sequence analysis (*Supplementary file 1*). Jumping spiders were fed about their own mass of leafhoppers *N. cincticeps* three times each week and examined 14–52 days following collection, as previously reported (*Taylor et al., 2014*).

### Electron microscopy

The Malpighian tubules of *N. cincticeps* were dissected using an astereomicroscope, serially fixed overnight at 4°C with 2.5% glutaraldehyde in 0.1 mol/L PBS, and post-fixed for 2 h at room temperature with 1% $OsO_4$. Then, samples were dehydrated in a series of ethanol solutions (50%, 70%, 80%, 90%, and 95%), then permeabilized with 100% ethanol and 100% acetone, followed by a series of epoxies in acetone (50%, 75%, and 100%). The pierced tissue was then immersed in epoxy resin and baked for more than 24 h at 70°C. Ultrathin slices were cut and stained with uranyl acetate for 15 min and lead citrate for 5 min, and examined by Hitachi H7800 microscope (Hitachi, Japan).

SEM was used to observe the morphology of leafhopper brochosomes. Samples were prepared as previously described (*Rakitov and Gorb, 2013a*). Leafhopper forewings were obtained by dissection under the stereomicroscope and glued onto SEM metal stubs with a double-sided carbon tape. The dried samples were coated with gold for 1 min in a MC1000 Ion Sputter Coater (Hitachi) and viewed with a HITACHI Regulus 8100 SEM (Hitachi).

### Spider predation experiment

Jumping spiders were starved for 3 days prior to the predation experiment to ensure that participating spiders were hungry enough to attack leafhoppers in our predation preference tests, but not so hungry that they would attack the first prey they encountered (without being choosy) (*Taylor et al., 2014*). For the jumping spiders to prey on, we placed two leafhoppers in a 9 cm diameter test arena (the bottom of the dish was covered with filter paper). Before starting the test, the spiders were placed in the arena for 15 min to adapt, and then they were released and permitted to prey on the leafhoppers. We directly observed the feeding process of the jumping spiders and recorded when the spiders attacked the first leafhopper and which leafhopper they preyed on. The predation finishes when the first leafhopper is attacked or when the jumping spider does not attack any leafhopper within 5 min. Each pair of jumping spider predation tests was independently replicated more than 60 times.

## Distribution of cuticle surface brochosomes and the optical properties of male and female insects at various periods post-eclosion

To investigate the distribution of brochosomes on the cuticle surface of adult *N. cincticeps* at various post-eclosion periods, we first collected female and male adult leafhoppers at 5, 10, 15, 20, and 25 days post-eclosion and dissected the forewings under a Nikon SMZ25 microscope (Nikon, Japan). The forewings were gold-sprayed and examined using HITACHI Regulus 8100 SEM (Hitachi). At each time point, 30 samples were randomly selected for SEM imaging. A 10 μm × 10 μm area within the SEM images was used to assess brochosome coverage, and ImageJ was employed for analysis (*Schneider et al., 2012*). Since brochosomes are synthesized at the distal section of the Malpighian tubule and the morphology of the Malpighian tubule in leafhoppers can also reflect brochosomes synthesis, we used Nikon SMZ25 microscope (Nikon) to examine the morphology of the Malpighian tubule in female and male adult leafhoppers at 5, 10, 15, 20, and 25 days post-eclosion.

To examine the optical properties of male and female insects at various periods post-eclosion, we first collected female and male adult leafhoppers at 5, 10, 15, 20, and 25 days post-eclosion, freeze-killed them, and placed them under white or UV light for observation through a Nikon SMZ25 microscope (Nikon). Then, the forewings of the leafhoppers were meticulously dissected, and the specular reflectance spectra of the forewings (25 × 25 μm) were precisely measured utilizing a 20/30 PV UV-Vis-NIR microspectrophotometer (CRAIC Technologies Inc, USA). UV-Vis-NIR reflectance spectra (250–1000 nm) were obtained using a ×10 UV-absorbing glass objective lens, with a minimum of five replicates per sample.

## Identification of brochosome structural protein-coding genes

Candidate brochosome structural protein-coding genes were identified through a combined analysis of transcriptomic and proteomic data. To knock down the expression of these candidates, RNAi was conducted by microinjecting double-stranded RNA (dsRNA) into the abdomens of fifth-instar nymphs. The dsRNAs targeting 500–1000 bp regions of Malpighian tubule-specific expression genes or GFP were synthesized in vitro using the T7 RiboMAX Express RNAi System (Promega, USA). The 50 fifth-instar nymphs of *N. cincticeps* were microinjected with 30 nl dsRNA (0.5 μg/μl) using the Nanoject II Auto-Nanoliter Injector (Drummond, USA). Thereafter, they were transferred to healthy rice seedlings for recovery. Forewings of treated leafhoppers were collected 7 days after microinjection (about 4–5 days post-eclosion), sprayed with gold, and analyzed with HITACHI Regulus 8100 SEM (Hitachi). Screening identified four brochosome structural protein-coding genes, and inhibiting their expression resulted in significant changes in brochosome morphology. The full-length sequence of these genes was obtained using by 5'-rapid amplification of cDNA ends (RACE) and 3'-RACE using SMARTer RACE 5'/3' Kit (Clontech, USA).

To analyze the temporal expression profiles of four brochosome structural protein-coding genes (GenBank accession numbers PP273097, PP273098, PP273099, PP273100), the whole body, salivary gland, midgut, ovary, and testis were dissected from 200 adult insects, total RNA in different tissue were extracted using TRIzol Reagent (Invitrogen, USA). The relative expression of these genes in different tissues of *N. cincticeps* adults was detected by RT-qPCR assay using a QuantStudio 5 Real-Time PCR system (Thermo Fisher Scientific, USA). The detected transcript levels were normalized to the transcript level of the housekeeping gene elongation factor 1 alpha (EF1α) (GenBank accession number AB836665) and estimated by the $2^{-\triangle\triangle Ct}$ (cycle threshold) method. Using the same procedure, the expression levels of these brochosome structural protein-coding genes were evaluated in female and male adult leafhoppers at 5, 10, 15, 20, and 25 days post-eclosion.

To investigate the effects of four BSM-encoding genes on brochosome morphology and distribution, dsRNAs were designed to target two nonoverlapping regions within the sequences of these genes. These dsRNAs were microinjected into the abdomens of leafhoppers both individually and in combination, with dsRNA targeting GFP serving as a negative control. Following a 7-day incubation period post-injection, changes in BSM gene expression were assessed in both control and treated groups via RT-qPCR. The morphology and distribution of brochosomes on the leafhopper cuticle were examined and documented using SEM. Additionally, ImageJ software was employed to quantify the proportion of morphologically abnormal brochosomes and the distribution area of brochosomes on the cuticle surface after injection of the dsRNA mixtures targeting the two nonoverlapping regions of

each BSM gene, as well as after dsGFP treatment. *Supplementary file 2* contains a list of the primers used in this study.

## Knocking down in vivo expression of BSM-coding genes in *N. cincticeps*

Following the prior procedure, dsRNA produced from four BSM-coding genes was injected into the abdomen of fifth-instar nymphs of *N. cincticeps* by microinjection, and a control group injected with dsGFP was established. After a 7-day incubation post-injection, alterations in the expression of the four BSM genes were monitored in the dsBSM and dsGFP-treated leafhoppers using RT-qPCR. The synthesis of brochosomes within the Malpighian tubules was investigated through TEM analysis. The morphology and distribution of brochosomes on the leafhopper cuticle were observed and imaged using SEM. To examine the optical properties, the dsBSM and dsGFP-treated leafhoppers, freeze-killed them, and placed them under white or UV light for observation through a Nikon SMZ25 microscope (Nikon). Subsequently, the forewings of the leafhoppers were meticulously dissected, and the specular reflectance spectra of the forewings (25 × 25 μm) were precisely measured utilizing a 20/30 PV UV-Vis-NIR microspectrophotometer (CRAIC Technologies, Inc). UV-Vis-NIR reflectance spectra (250–1000 nm) were obtained using a ×10 UV-absorbing glass objective lens, with a minimum of five replicates per sample. Finally, the predation preference of the jumping spiders on the dsBSM and dsGFP treated leafhoppers was examined by a predation experiment.

## In vitro measurements were conducted to assess the relationship between brochosome morphology and their optical performance

To collect approximately 500 *N. cincticeps* treated with dsGFP or dsBSM, dissect the forewings of the leafhoppers and immerse them in acetone in a 50 ml centrifuge tube. Place the centrifuge tubes on a room temperature orbital shaker at 50 rpm for 12 h to separate the forewings from the leafhoppers due to friction. Following this, centrifuge at 1000 × *g* for 10 min to separate the brochosomes. Resuspend the particles in fresh acetone, briefly sonicate, then centrifuge at 1000 × *g* for 10 min. Repeat this process three times. Resuspend the collected brochosomes from dsGFP and dsBSM-treated leafhopper forewings in acetone and adjust the two suspension solutions to the same OD280 reading. Next, using a pipette, carefully drop the brochosome solution onto quartz slides and brown planthopper wings. Allow the acetone to completely evaporate at room temperature. At the same time, we also set up quartz slides and brown planthopper wings that were treated with acetone alone as negative controls. Prepared quartz slides and brown planthopper wings can be observed under SEM to examine the distribution of brochosomes. Specular reflectance measurements of quartz substrates and brown planthopper wing sections (10 × 10 μm) were performed using a 20/30 PV UV-Vis-NIR microspectrophotometer (CRAIC Technologies, Inc). Reflectance spectra were systematically acquired across the 250–1000 nm spectral range, with a minimum of five replicates per sample.

To investigate the correlation between BSM proteins and the optical characteristics of brochosomes, the four genes encoding BSM proteins were separately cloned into pET-28a or pGEX-4T2 vectors. Following induction, the expressed proteins were purified from cell lysates via batch affinity chromatography using either glutathione (GSH) agarose or nickel-nitrilotriacetic acid (Ni-NTA) agarose, depending on the fusion tag. GST protein was produced following the identical protocol from cells harboring the empty pGEX-4T2 vector. The purified BSM fusion proteins and GST protein were normalized to an equivalent OD280 value. These protein solutions were then meticulously spotted onto quartz slides and permitted to dry completely at ambient temperature. In parallel, quartz slides treated exclusively with PBS served as negative controls. UV-Vis-NIR reflectance spectra (250–1000 nm) of quartz substrates (10 × 10 μm) were obtained using a 20/30 PV UV-Vis-NIR microspectrophotometer (CRAIC Technologies, Inc), with a minimum of five replicates per sample.

## Removal of leafhopper forewing brochosome by acetone

Previous research has shown that acetone may efficiently remove brochosomes from the cuticle surface of leafhoppers (*Rakitov et al., 2018*). We manually removed the brochosomes from leafhopper forewings using acetone and subsequently examined the distribution of brochosomes on the forewings using SEM to confirm the removal efficacy of acetone. Imaging observations were conducted under white or UV light through a Nikon SMZ25 microscope (Nikon). The specular reflection of the leafhopper

forewing region (25 × 25 μm) was measured using a 20/30 PV UV-Vis-NIR microspectrophotometer (CRAIC Technologies, Inc) across 250–1000 nm, with a minimum of five replicates per sample.

## Bioinformatics analysis and phylogenetic tree

The brochosome is a unique secretion produced by leafhoppers (Cicadellidae). In order to determine the distribution of brochosome structural protein-coding genes in Hemiptera, the brochosome structural protein-coding genes were used as queries to search for homologous sequences in Hemiptera transcriptomes. We examined 116 Hemiptera species, encompassing major families (Acanaloniidae, Achilidae, Aetalionidae, Aphrophoridae, Caliscelidae, Cercopidae, Cicadellidae, Cicadidae, Cixiidae, Clastopteridae, Delphacidae, Derbidae, Dictyopharidae, Epipygidae, Eurybrachidae, Flatidae, Fulgoridae, Issidae, Machaerotidae, Melizoderidae, Membracidae, Myerslopiidae, Nogodinidae, Peloridiidae, Ricaniidae, Tettigarctidae, Tettigometridae, Theaceae, Tropiduchidae). Transcriptome data of 116 Hemiptera species were downloaded from the NCBI Sequence Read Archive (*Supplementary file 2*) and assembled using SOAPdenovo-Trans (version 1.01) (*Xie et al., 2014*). The resulting transcripts were filtered to remove potential contaminants. We compared four identified BSM-encoded genes with these transcriptomes using the tblastn (E-value < 1.0e–5). Heatmaps were generated using TBtools software (*Chen et al., 2023*). At least one homologous gene for the four BSM genes was identified in 66 species. A species phylogenetic tree for these 66 Hemiptera species was constructed following the methods outlined in previous studies (*Johnson et al., 2018*). OrthoFinder was used to identify orthogroups and infer the species tree, obtaining 1683 single-copy orthologous genes after filtering (*Emms and Kelly, 2019*). MAFFT was employed for sequence alignment, followed by trimming with trimAI (*Capella-Gutiérrez et al., 2009*; *Katoh and Standley, 2013*). The best amino acid substitution model was selected using ModelFinder based on the Bayesian Information Criterion (*Kalyaanamoorthy et al., 2017*). The maximum likelihood phylogenetic tree was constructed using IQ-TREE v1.6.8 with 1000 ultrafast bootstrap replicates (*Minh et al., 2020*). All analyses were conducted using PhyloSuite v1.2.2, which integrates MAFFT, trimAI, ModelFinder, and IQ-TREE into a unified pipeline (*Capella-Gutiérrez et al., 2009*; *Emms and Kelly, 2019*; *Kalyaanamoorthy et al., 2017*; *Katoh and Standley, 2013*; *Minh et al., 2020*; *Zhang et al., 2020*).

The amino acid sequences of BSM-2 and BSM-3 were aligned using the Clustal W multiple sequence alignment program (*Larkin et al., 2007*). AlphaFold2 was employed to predict the three-dimensional structures of BSM-2 and BSM-3, and structural alignments were conducted using PyMOL (*DeLano, 2002*; *Jumper et al., 2021*). In Membracoidea, the maximum likelihood phylogenetic tree for BSM1-4 was constructed using MAFFT, trimAI, ModelFinder, and IQ-TREE in PhyloSuite (*Capella-Gutiérrez et al., 2009*; *Emms and Kelly, 2019*; *Kalyaanamoorthy et al., 2017*; *Katoh and Standley, 2013*; *Minh et al., 2020*; *Xiang et al., 2023*; *Zhang et al., 2020*).

## Statistical analyses

All experiments were performed at least three independent replicates, and statistical analyses were performed with GraphPad Prism8.0 software. Statistical significance was calculated by a two-tailed Student's *t*-test, one-way ANOVA, two-way ANOVA, and/or unpaired Student's *t*-test. p-values<0.05 were considered statistically significant.

## Acknowledgements

This work received support from the National Natural Science Foundation of China (U23A6006, U20A2036, 32270150) and the Natural Science Foundation of Ningbo Municipality (2023J112). We acknowledge Dr. Yuan Cheng from the Instrumentation and Service Center for Molecular Science (ISCMS) at Westlake University for assistance with optical measurement and data interpretation. We are grateful for the support provided by the Bioimaging Center of the Bioimaging Center, State Key Laboratory of Agricultural Products Safety, Institute of Plant Virology, Ningbo University, in TEM and SEM measurement and data interpretation. Additionally, we thank Professor Shou-Wei Ding (University of California, Riverside, CA, USA) for his valuable and constructive suggestions for manuscript improvement.

## Additional information

### Funding

| Funder | Grant reference number | Author |
|---|---|---|
| National Natural Science Foundation of China | U23A6006 | Jian-Ping Chen |
| National Natural Science Foundation of China | U20A2036 | Jian-Ping Chen |
| National Natural Science Foundation of China | 32270150 | Qianzhuo Mao |
| Natural Science Foundation of Ningbo Municipality | 2023J112 | Wei Wu |

The funders had no role in study design, data collection and interpretation, or the decision to submit the work for publication.

### Author contributions

Wei Wu, Conceptualization, Data curation, Investigation, Methodology, Writing – original draft, Writing – review and editing; Qianzhuo Mao, Conceptualization, Visualization; Zhuang-Xin Ye, Data curation, Software; Zhenfeng Liao, Visualization; Hong-Wei Shan, Writing – original draft; Jun-Min Li, Conceptualization, Writing – original draft, Writing – review and editing; Chuan-Xi Zhang, Conceptualization, Writing – review and editing; Jian-Ping Chen, Conceptualization, Funding acquisition, Writing – original draft, Project administration, Writing – review and editing

### Author ORCIDs

Wei Wu ⬥ https://orcid.org/0000-0002-5440-345X

Reviewer #1 (Public review): https://doi.org/10.7554/eLife.99639.3.sa1
Reviewer #2 (Public review): https://doi.org/10.7554/eLife.99639.3.sa2
Author response https://doi.org/10.7554/eLife.99639.3.sa3

## Additional files

### Supplementary files

Supplementary file 1. List of oligonucleotide primers used in this study.

Supplementary file 2. Information on Hemipteran insects used in the search for homologs of BSM.

MDAR checklist

### Data availability

All the data needed to understand and assess the conclusions of this research are available in the article, Supplementary Materials, and GenBank (accession number PP273097, PP273098, PP273099, PP273100). Any additional information required to reanalyse the data reported in this paper is available from the lead contact upon request.

The following datasets were generated:

| Author(s) | Year | Dataset title | Dataset URL | Database and Identifier |
|---|---|---|---|---|
| Wu W | 2024 | Nephotettix cincticeps brochosomin-1 (BSM-1) mRNA, complete cds | https://www.ncbi.nlm.nih.gov/nuccore/PP273097.1/ | NCBI GenBank, PP273097 |

*Continued on next page*

*Continued*

| Author(s) | Year | Dataset title | Dataset URL | Database and Identifier |
|-----------|------|---------------|-------------|-------------------------|
| Wu W | 2024 | Nephotettix cincticeps brochosomin-2 (BSM-2) mRNA, complete cds | https://www.ncbi.nlm.nih.gov/nuccore/PP273098.1/ | NCBI GenBank, PP273098 |
| Wu W | 2024 | Nephotettix cincticeps brochosomin-3 (BSM-3) mRNA, complete cds | https://www.ncbi.nlm.nih.gov/nuccore/PP273099.1/ | NCBI GenBank, PP273099 |
| Wu W | 2024 | Nephotettix cincticeps brochosomin-4 (BSM-4) mRNA, complete cds | https://www.ncbi.nlm.nih.gov/nuccore/PP273100.1/ | NCBI GenBank, PP273100 |

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
