## [Editor Report · eLife Assessment]

The authors provide **important** insights into a system of insect camouflage where a coating of self-made nanoparticles (brochosomes) reduces the reflection of UV light, leading to lower predation by spiders. **Compelling** evidence is provided by micro-UV-Vis spectroscopy, electron microscopy, transcriptome and proteome analysis, histology, in vivo predation assays, and gene knockdowns. The phylogenetic analyses provide evidence that the genes coding for the brochosome proteins are clade-specific and have diversified by gene duplication.

---

## [Referee Report · Reviewer #1 (Public review)]

Summary:

Evading predation is of utmost importance for most animals and camouflage is one of the predominant mechanisms. Wu et al. set out to test the hypothesis of a unique camouflage system in leafhoppers. These animals coat themselves with brochosomes, which are spherical nanostructures that are produced in the Malpighian tubules and are distributed on the cuticle after eclosion. Based on previous findings on reflectivity properties of brochosomes, the authors provide convincing evidence that these nanostructures indeed reduce reflectivity of the animals thereby reducing predation by jumping spiders. Further, they identify four proteins, which are essential for proper development and function of brochosomes: In RNAi experiments, the regular brochosome structure is lost, the reflectivity reduced and the respective animals are prone to increased predation. Finally, the authors provide phylogenetic sequence analyses and speculate about the evolution of these genes.

Strengths:

The study is very comprehensive including careful optical measurements, EM and TM analysis of the nanoparticles and their production line in the malphigian tubules, in vivo predation tests and knock-down experiments to identify essential proteins. Indeed, the results are very convincingly in line with the starting hypothesis such that the study robustly assigns a new biological function to the brochosome coating system.

A key strength of the study is that the biological relevance of the brochosome coating is convincingly shown by an in vivo predation test using a known predator from the same habitat.

Another major step forward is an RNAi screen, which identified four proteins, which are essential for the brochosome structure (BSMs). After respective RNAi knock-downs, the brochosomes show curious malformations that are interesting in terms of the self-assembly of these nanostructures. The optical and in vivo predation tests provide excellent support for the model that the RNAi knock-down leads to a change of brochosomes structure, which reduces reflectivity, which in turn leads to a decrease of the antipredatory effect.

Conclusion:

The authors successfully tested their hypothesis in a multidisciplinary approach and convincingly assigned a new biological function to the brochosomes system. The results fully support their claims on the involvement of the four BSM genes in brochosome structure, the relevance of brochosomes for predation avoidance and they provide evidence for the evolution of these genes.

The work is a very interesting study case of the evolutionary emergence of a new system to evade predators. Based on this study, the function of the BSM genes could now be studied in other species to provide insights into putative ancestral functions. Further, studying the self-assembly of such highly regular complex nano-structures will be strongly fostered by the identification of the four key structural genes.

---

## [Referee Report · Reviewer #2 (Public review)]

Summary:

In this manuscript, the authors investigate the optical properties of brochosomes produced by leafhoppers. They hypothesize that brochosomes reduce light reflection on the leafhopper's body surface, aiding in predator avoidance. Their hypothesis is supported by experiments involving jumping spiders. Additionally, the authors employ a variety of techniques including micro-UV-Vis spectroscopy, electron microscopy, transcriptome and proteome analysis, and bioassays. This study is highly interesting, and the experimental data is well-organized and logically presented.

Strengths:

The use of brochosomes as a camouflage coating has been hypothesized since 1936 (R.B. Swain, Entomol. News 47, 264-266, 1936) with evidence demonstrated by similar synthetic brochosome systems in a number of recent studies (S. Yang, et al. Nat. Commun. 8:1285, 2017; L. Wang, et al., PNAS. 121: e2312700121, 2024). However, direct biological evidence or relevant field studies have been lacking to directly support the hypothesis that brochosomes are used for camouflage. This work provides the first biological evidence demonstrating that natural brochosomes can be used as a camouflage coating to reduce the leafhoppers' observability to their predators. The design of the experiments is novel.

Weaknesses:

(1) The observation that brochosome coatings become sparse after 25 days in both male and female leafhoppers, resulting in increased predation by jumping spiders, is intriguing. However, since leafhoppers consistently secrete and groom brochosomes, it would be beneficial to explore why brochosomes become significantly less dense after 25 days.

(2) The authors demonstrate that brochosome coatings reduce UV (specular) reflection compared to surfaces without brochosomes, which can be attributed to the rough geometry of brochosomes as discussed in the literature. However, it would be valuable to investigate whether the proteins forming the brochosomes are also UV absorbing.

(3) The experiments with jumping spiders show that brochosomes help leafhoppers avoid predators to some extent. It would be beneficial for the authors to elaborate on the exact mechanism behind this camouflage effect. Specifically, why does reduced UV reflection aid in predator avoidance? If predators are sensitive to UV light, how does the reduced UV reflectance specifically contribute to evasion?

(4) An important reference regarding the moth-eye effect is missing. Please consider including the following paper: Clapham, P. B., and M. C. Hutley. "Reduction of lens reflection by the 'Moth Eye' principle." Nature 244: 281-282 (1973).

(5) The introduction should be revised to accurately reflect the related contributions in literature. Specifically, the novelty of this work lies in the demonstration of the camouflage effect of brochosomes using jumping spiders, which is verified for the first time in leafhoppers. However, the proposed use of brochosome powder for camouflage was first described by R.B. Swain (R.B. Swain, Notes on the oviposition and life history of the leafhopper Oncometopta undata Fabr. (Homoptera: Cicadellidae), Entomol. News. 47: 264-266 (1936)). Recently, the antireflective and potential camouflage functions of brochosomes were further studied by Yang et al. based on synthetic brochosomes and simulated vision techniques (S. Yang, et al. "Ultra-antireflective synthetic brochosomes." Nature Communications 8: 1285 (2017)). Later, Lei et al. demonstrated the antireflective properties of natural brochosomes in 2020 (C.-W. Lei, et al., "Leafhopper wing-inspired broadband omnidirectional antireflective embroidered ball-like structure arrays using a nonlithography-based methodology." Langmuir 36: 5296-5302 (2020)). Very recently, Wang et al. successfully fabricated synthetic brochosomes with precise geometry akin to those natural ones, and further elucidated the antireflective mechanisms based on the brochosome geometry and their role in reducing the observability of leafhoppers to their predators (L. Wang et al. "Geometric design of antireflective leafhopper brochosomes." Proceedings of the National Academy of Sciences 121: e2312700121 (2024)).

Comments on revisions:

In this revision, the authors have addressed some of the key concerns I raised in our previous comments. However, a few issues remain unaddressed. Additionally, the new experimental data introduced in the manuscript require further clarification, which I outline below.

(1) As I pointed out in my previous review comments, "The use of brochosomes as a camouflage coating has been hypothesized since 1936 (R.B. Swain, Entomol. News 47, 264-266, 1936) with evidence demonstrated by similar synthetic brochosome systems in a number of recent studies (S. Yang, et al. Nat. Commun. 8:1285, 2017; L. Wang, et al., PNAS. 121: e2312700121, 2024). However, direct biological evidence or relevant field studies have been lacking to directly support the hypothesis that brochosomes are used for camouflage." While the authors did cite the original hypothesis proposed by R.B. Swain (1936), they have omitted important references that provide evidence on the use of antireflective properties of brochosomes for camouflage in a synthetic setting (see for example, Fig. 5a of S. Yang, et al. Nat. Commun. 8:1285, 2017). The authors are recommended to revise the Abstract and Introduction accordingly to ensure a fair and accurate representation of the existing literature.

(2) The antireflection mechanisms of brochosome structures have been discussed in detail, specifically, how their geometries (i.e., brochosome diameter and pore size) contribute to reducing UV reflectance (L. Wang, et al., PNAS. 121: e2312700121, 2024 and P. Banergee, et al., Advanced Photonics Research 4:2200343, 2023). The authors should incorporate these recent findings into their discussion (line 381 - line 383 of the manuscript).

(3) The authors presented new data brochosomes deposited on a quartz slide and measured their reflectance across UV, visible light, and infrared wavelengths. Since reflectance is highly sensitive to the uniformity of brochosome coverage on the substrate, it is crucial to quantify this coverage across the measurement area for comparison. While the authors include SEM images to illustrate the packing of brochosomes on both the leafhopper wing and the quartz substrate (Fig. S7) at a microscopic scale (~10 um view), it would be beneficial to also provide SEM images at a larger scale (e.g., 100 um - 1 mm) and quantify the density of brochosomes per unit area for comparison.

(4) For the negative control using acetone to remove the brochosomes the leafhopper wing, have the authors confirmed the absence of brochosomes after treatment? If so, the authors should explicitly indicate this for clarity.

---

## [Author Response]

The following is the authors’ response to the original reviews

**Reviewer #1 (Public review):**
Summary:Evading predation is of utmost importance for most animals and camouflage is one of the predominant mechanisms. Wu et al. set out to test the hypothesis of a unique camouflage system in leafhoppers. These animals coat themselves with brochosomes, which are spherical nanostructures that are produced in the Malpighian tubules and are distributed on the cuticle after eclosion. Based on previous findings on the reflectivity properties of brochosomes, the authors provide very good evidence that these nanostructures indeed reduce the reflectivity of the animals thereby reducing predation by jumping spiders. Further, they identify four proteins, which are essential for the proper development and function of brochosomes. In RNAi experiments, the regular brochosome structure is lost, the reflectivity reduced and the respective animals are prone to increased predation. Finally, the authors provide some phylogenetic sequence analyses and speculate about the evolution of these essential genes.Strengths:The study is very comprehensive including careful optical measurements, EM and TM analysis of the nanoparticles and their production line in the malphigian tubules, in vivo predation tests, and knock-down experiments to identify essential proteins. Indeed, the results are very convincingly in line with the starting hypothesis such that the study robustly assigns a new biological function to the brochosome coating system.A key strength of the study is that the biological relevance of the brochosome coating is convincingly shown by an in vivo predation test using a known predator from the same habitat.Another major step forward is an RNAi screen, which identified four proteins, which are essential for the brochosome structure (BSMs). After respective RNAi knock-downs, the brochosomes show curious malformations that are interesting in terms of the self-assembly of these nanostructures. The optical and in vivo predation tests provide excellent support for the model that the RNAi knock-down leads to a change of brochosomes structure, which reduces reflectivity, which in turn leads to a decrease of the antipredatory effect.

Thank you very much for your positive feedback and insightful comments on our manuscript. We are delighted that you acknowledge the efforts we have made in studying the components and functions of Brochosomal proteins. We have carefully considered your suggestions and have thoroughly revised the manuscript to address the shortcomings identified in our original submission. We hope that the revised version meets with your approval. Below, please find our detailed point-by-point responses.

Weaknesses:The reduction of reflectivity by aberrant brochosomes or after ageing is only around 10%. This may seem little to have an effect in real life. On the other hand, the in vivo predation tests confirm an influence. Hence, this is not a real weakness of the study - just a note to reconsider the wording for describing the degree of reflectivity.

Thank you for your valuable suggestions. Based on your recommendations, we have revised the manuscript accordingly. Although the absolute reduction in light reflection due to Brochosomal coverage is approximately 10%, the relative decrease in light reflection on the leafhopper's surface is nearly 30%. Specifically, in the ultraviolet region, the reflection is reduced from about 30% to 20%, and in the visible light region, it is reduced from 20% to 10%. For detailed revisions, please refer to lines 151-156 of the revised manuscript.

The single gene knockdowns seemed to lead to a very low penetrance of malformed brochosomes (Figure Supplement 3). Judging from the overview slides, less than 1% of brochosomes may have been affected. A quantification of regular versus abnormal particles in both, wildtype and RNAi treatments would have helped to exclude that the shown aberrant brochosomes did not just reflect a putative level of "normal" background defects. Of note, the quadruple knock-down of all BSMs seemed to lead to a high penetrance (Figure 4), which was already reflected in the microtubule production line. While the data shown are convincing, a quantification might strengthen the argument.While the RNAi effects seemed to be very specific to brochosomes and therefore very likely specific, an off-target control for RNAi was still missing. Finding the same/similar phenotype with a non-overlapping dsRNA fragment in one off-target experiment is usually considered required and sufficient. Further, the details of the targeted sequence will help future workers on the topic.

Thank you for your valuable suggestions. Based on your recommendations, we have synthesized dsRNA targeting two non-overlapping regions of the coding sequences for four Brochosomal structural protein genes. These dsRNAs were injected individually and in combination for each gene. Our RNAi experiments for each BSM gene demonstrated that both individual and combined injections significantly suppressed the expression of the target genes, with the combined injection yielding slightly better silencing efficiency. Statistical analysis of the SEM observations revealed that the combined injection of dsRNAs targeting two non-overlapping regions led to a 60-70% reduction in the surface area coverage of Brochosomes. Additionally, approximately 20% of the remaining Brochosomes exhibited significant morphological changes. For detailed revisions, please refer to lines 199-211 of the revised manuscript, as well as Figures 3A and 3C, and Supplementary Figures 4 and 5.

The main weakness in the current manuscript may be the phylogenetic analysis and the model of how the genes evolved. Several aspects were not clearly or consistently stated such that I felt unsure about what the authors actually think. For instance: Are all the 4 BSMs related to each other or only BSM2 and 3? If so, not only BSM2 and 3 would be called "paralogs" but also the other BSMs. If they were all related, then a phylogenetic tree including all BSMs should be shown to visualize the relatedness (including the putative ancestral gene if that is the model of the authors). Actually, I was not sure about how the authors think about the emergence of the BSMs. Are they real orphan genes (i.e. not present outside the respective clade) or was there an ancestral gene that was duplicated and diverged to form the BSMs? Where in the phylogeny does the first of the BSMs or ancestral proteins emerge (is the gene found in Clastoptera arizonana the most ancestral one?)? Maybe, the evolution of the BSMs would have to be discussed individually for each gene as they show somewhat different patterns of emergence and loss (BSM-4 present in all species, the others with different degrees of phylogenetic restriction).

Thank you very much for your constructive feedback on our phylogenetic analysis and the modeling of gene evolution. We fully agree with your insights and acknowledge that the evolutionary analysis of BSM genes remains somewhat ambiguous. This ambiguity is primarily due to the limited research on the precise structural protein composition of Brochosomes. While proteomics studies have analyzed and discussed the structural proteins of Brochosomes, the accurate composition of these proteins is still poorly understood. In this study, we identified four BSM proteins, but given the intricate structure of Brochosomes as proteinaceous spheres, we believe there may be additional BSM genes that have not yet been identified. Moreover, despite the presence of over ten thousand species within the Cicadomorpha, only three species have genome sequences available, and fewer than a hundred species have transcriptome sequencing data. The scarcity of research on Brochosomes, as well as the limited availability of genomic and transcriptomic data, poses significant challenges for our phylogenetic analysis and understanding of BSM gene evolution.

Based on your suggestions, we have revised the manuscript accordingly. Specifically, we have updated Figure 5C by including ten additional species from Cereopoidea, Cicadoidea, and Fulgoroidea to better illustrate that BSM genes are true orphan genes. We have also added a phylogenetic tree of BSM genes within Cicadidae in Supplementary Figure 3. Additionally, we have expanded the discussion of BSM gene evolution in the manuscript (lines 503-556). For detailed revisions, please refer to Figure 5C, Supplementary Figure 3, and lines 507-585 of the revised manuscript.

Related to these questions I remained unsure about some details in Figure 5. On what kind of analysis is the phylogeny based? Why are some species not colored, although they are located on the same branch as colored ones? What is the measure for homology values - % identity/similarity? The homology labels for Nephotetix cincticeps and N. virescens seem to be flipped: the latter is displayed with 100% identity for all genes with all proteins while the former should actually show this. As a consequence of these uncertainties, I could not fully follow the respective discussion and model for gene evolution.

Thank you very much for your insightful comments and suggestions. We have carefully considered your feedback and have thoroughly revised our manuscript accordingly. Specifically, we have enhanced the description of the phylogenetic analysis process to provide greater clarity and transparency, with the detailed methods now included in lines 789-798. Regarding Figure 5C, we appreciate your attention to the coloring scheme. We would like to clarify that the family Cicadellidae comprises 25 subfamilies, many of which are represented by only one species in our figure. To ensure clarity and meaningful representation, we have chosen to color only those subfamilies with more than three species, thereby avoiding visual clutter and emphasizing the most relevant taxonomic groups. Additionally, we have corrected the inverted homology labels for *Nephotetix cincticeps* and *Nephotetix virescens* to ensure the accuracy and consistency of our data presentation.

Conclusion:The authors successfully tested their hypothesis in a multidisciplinary approach and convincingly assigned a new biological function to the brochosomes system. The results fully support their claims - only the quantification of the penetrance in the RNAi experiments would be helpful to strengthen the point. The author's analysis of the evolution of BSM genes remained a bit vague and I remained unsure about their respective conclusions.The work is a very interesting study case of the evolutionary emergence of a new system to evade predators. Based on this study, the function of the BSM genes could now be studied in other species to provide insights into putative ancestral functions. Further, studying the self-assembly of such highly regular complex nano-structures will be strongly fostered by the identification of the four key structural genes.
**Reviewer #1 (Recommendations for the authors):**
Main manuscript:Please consider the annotated pdf with suggestions for wording and comments at the authors' discretion:

Thank you very much for your detailed suggestions and comments provided in the annotated PDF. We have carefully reviewed each of your points and have revised the manuscript accordingly. All changes have been highlighted in red text for your convenience. The revised manuscript with tracked changes is available for your review. We believe these revisions have improved the clarity and quality of our manuscript. Thank you again for your valuable feedback.

Supplementary Figure 2 C:Y-axes:- label: "surface coverage in %"- there are different scale values for the different days (e.g. 80-105 for day 5 and 0-80 at day 25). As a comparison between days is interesting, it would help to have the same scale values for all. That would show the decrease more intuitively.

Thank you very much for your suggestion regarding the Y-axis in Supplementary Figure 2C. We agree that using a consistent scale across all time points is essential for clear and intuitive comparison. In the revised manuscript, we have standardized the Y-axis scale for Supplementary Figure 2C to a uniform range of 0-100% for all days. This change allows for a more straightforward visualization of the decreasing trend in surface coverage over time.

**Reviewer #3 (Public review):**
Summary:In this manuscript, the authors investigate the optical properties of brochosomes produced by leafhoppers. They hypothesize that brochosomes reduce light reflection on the leafhopper's body surface, aiding in predator avoidance. Their hypothesis is supported by experiments involving jumping spiders. Additionally, the authors employ a variety of techniques including micro-UV-Vis spectroscopy, electron microscopy, transcriptome and proteome analysis, and bioassays. This study is highly interesting, and the experimental data is well-organized and logically presented.Strengths:The use of brochosomes as a camouflage coating has been hypothesized since 1936 (R.B. Swain, Entomol. News 47, 264-266, 1936) with evidence demonstrated by similar synthetic brochosome systems in a number of recent studies (S. Yang, et al. Nat. Commun. 8:1285, 2017; L. Wang, et al., PNAS. 121: e2312700121, 2024). However, direct biological evidence or relevant field studies have been lacking to directly support the hypothesis that brochosomes are used for camouflage. This work provides the first biological evidence demonstrating that natural brochosomes can be used as a camouflage coating to reduce the leafhoppers' observability of their predators. The design of the experiments is novel.

We are extremely grateful for your positive feedback and insightful comments on our manuscript. We are delighted that you have recognized the efforts we have put into our research on how brochosomes serve as a camouflage coating to reduce the detectability of leafhoppers to their predators. We have carefully considered your suggestions and have thoroughly revised the manuscript to address the shortcomings of the original version. We hope that the revised version meets with your approval. Below, please find our detailed point-by-point responses.

Weaknesses:(1) The observation that brochosome coatings become sparse after 25 days in both male and female leafhoppers, resulting in increased predation by jumping spiders, is intriguing. However, since leafhoppers consistently secrete and groom brochosomes, it would be beneficial to explore why brochosomes become significantly less dense after 25 days.

Thank you very much for your valuable suggestions. We appreciate your interest in the reduction of brochosomal density on the surface of leafhoppers after 25 days.We believe that the primary reason for the decreased density of brochosomes on the leafhopper surface after 25 days is the reduced synthesis and secretion of brochosomes. The Malpighian tubules are the main sites for brochosome synthesis. As shown in Figure 2D and Supplementary Figure 1, the thick glandular segments of the Malpighian tubules in both male and female leafhoppers begin to atrophy 15 days after reaching adulthood. This indicates a gradual decline in brochosome synthesis and secretion after day 15 of adulthood. Following your suggestion, we have revised the discussion section of the manuscript to elaborate on this observation. The detailed changes can be found in lines 474-491 of the revised manuscript.

(2) The authors demonstrate that brochosome coatings reduce UV (specular) reflection compared to surfaces without brochosomes, which can be attributed to the rough geometry of brochosomes as discussed in the literature. However, it would be valuable to investigate whether the proteins forming the brochosomes are also UV absorbing.

Thank you very much for your valuable suggestions. Following your advice, we have successfully expressed four BSM genes in a prokaryotic system, purified the corresponding proteins, and applied them to quartz glass surfaces. We then measured the light reflectance of the quartz glass surfaces coated with these purified proteins. The results showed that the purified BSM proteins did not exhibit better antireflective properties compared to the control GST protein. For more details, please refer to Supplementary Figure 8 in the revised manuscript. We believe that the excellent antireflective properties of brochosomes are fundamentally due to their unique geometric shapes. The hollow pores within the brochosomes, with diameters of approximately 100 nm, are significantly smaller than most wavelengths in the visible spectrum. When light passes through these tiny pores, diffraction occurs, while light passing through the ridges of the brochosomes causes scattering. The interference between the diffracted and scattered light from these pores and ridges results in the observed extinction characteristics of brochosomes. We have incorporated these insights into the discussion section of the revised manuscript (lines 416-425 and lines 432-442 of the revised manuscript).

(3) The experiments with jumping spiders show that brochosomes help leafhoppers avoid predators to some extent. It would be beneficial for the authors to elaborate on the exact mechanism behind this camouflage effect. Specifically, why does reduced UV reflection aid in predator avoidance? If predators are sensitive to UV light, how does the reduced UV reflectance specifically contribute to evasion?

Thank you very much for your valuable suggestions. Based on your advice, we have included a detailed discussion on how reducing ultraviolet (UV) reflection can help insects avoid predation. The revised content can be found in lines 445-460 of the revised manuscript.

“UV light serves as a crucial visual cue for various insect predators, enhancing foraging, navigation, mating behavior, and prey identification (Cronin & Bok, 2016; Morehouse et al., 2017; Silberglied, 1979). Predators such as birds, reptiles, and predatory arthropods often rely on UV vision to detect prey (Church et al., 1998; Li & Lim, 2005; Zou et al., 2011). However, UV reflectance from insect cuticles can disrupt camouflage, increasing the risk of detection and predation, as natural backgrounds like leaves, bark, and soil typically reflect minimal UV light (Endler, 1997; Li & Lim, 2005; Tovee, 1995). To mitigate this risk, insects often possess anti-reflective cuticular structures that reduce UV and broad-spectrum light reflectance. This strategy is widespread among insects, including cicadas, dragonflies, and butterflies, and has been shown to decrease predator detection rates (Hooper et al., 2006; Siddique et al., 2015; Zhang et al., 2006). For example, the compound eyes of moths feature hexagonal protuberances that reduce UV reflectance, aiding nocturnal concealment (Blagodatski et al., 2015; Stavenga et al., 2005). In butterflies, UV reflectance from eyespots on wings can attract predators, but reducing UV reflectance or eyespot size can lower predation risk and enhance camouflage (Chan et al., 2019; Lyytinen et al., 2004). Hence, the reflection of ultraviolet light from the insect cuticle surface increases the risk of predation by disrupting camouflage (Tovee, 1995)”

(4) An important reference regarding the moth-eye effect is missing. Please consider including the following paper: Clapham, P. B., and M. C. Hutley. "Reduction of lens reflection by the 'Moth Eye' principle." Nature 244: 281-282 (1973).

Thank you very much for pointing out the omission of the important reference on the “moth eye” effect. We sincerely apologize for the oversight. Based on your suggestion, we have now included the seminal paper by Clapham and Hutley (1973) in the revised manuscript. The reference has been added to both the Introduction and Discussion sections to provide a more comprehensive context for our discussion on anti-reflective structures in insects.

(5) The introduction should be revised to accurately reflect the related contributions in literature. Specifically, the novelty of this work lies in the demonstration of the camouflage effect of brochosomes using jumping spiders, which is verified for the first time in leafhoppers. However, the proposed use of brochosome powder for camouflage was first described by R.B. Swain (R.B. Swain, Notes on the oviposition and life history of the leafhopper Oncometopta undata Fabr. (Homoptera: Cicadellidae), Entomol. News. 47: 264-266 (1936)). Recently, the antireflective and potential camouflage functions of brochosomes were further studied by Yang et al. based on synthetic brochosomes and simulated vision techniques (S. Yang, et al. "Ultra-antireflective synthetic brochosomes." Nature Communications 8: 1285 (2017)). Later, Lei et al. demonstrated the antireflective properties of natural brochosomes in 2020 (C.-W. Lei, et al., "Leafhopper wing-inspired broadband omnidirectional antireflective embroidered ball-like structure arrays using a nonlithography-based methodology." Langmuir 36: 5296-5302 (2020)). Very recently, Wang et al. successfully fabricated synthetic brochosomes with precise geometry akin to those natural ones, and further elucidated the antireflective mechanisms based on the brochosome geometry and their role in reducing the observability of leafhoppers to their predators (L. Wang et al. "Geometric design of antireflective leafhopper brochosomes." Proceedings of the National Academy of Sciences 121: e2312700121 (2024)).

Thank you very much for your valuable suggestions regarding the revision of the introduction to accurately reflect the relevant contributions in the literature. Based on your feedback, we have thoroughly revised the introduction and added the suggested references to provide a comprehensive context for our study. The details of these revisions can be found in lines 84-94 of the revised manuscript.

**Reviewer #3 (Recommendations for the authors):**
(1) In Figure 2E, the data for Male-5d appears to be missing. Please verify and ensure all relevant data is included.

Thank you for pointing out the issue regarding the data presentation in Figure 2E.We apologize for any confusion caused by the overlapping data points and the less conspicuous color choice for Male-5d. We have carefully reviewed the data and confirmed that all relevant data points, including Male-5d, are indeed present in the dataset. In the revised manuscript, we have adjusted the color scheme for Male-5d and Female-5d in Figure 2E to ensure that both curves are clearly distinguishable, even in areas where they overlap. This adjustment should facilitate a more accurate and convenient observation of the data trends. We appreciate your attention to detail, and we believe these revisions have improved the clarity and readability of the figure.

(2) In Figure 6, please clarify the reflectance data in the inset. Clearly explain what the blue and light blue curves represent.

Thank you for your suggestion regarding Figure 6.We have revised the figure to improve clarity. The light blue curve now represents the reflectance measurements of leafhoppers with higher brochosome coverage, while the dark blue curve corresponds to those with lower coverage. These changes, along with updated labels in the figure legend, ensure that the data are clearly distinguishable and easy to interpret. We appreciate your feedback and believe these revisions have enhanced the overall clarity of the figure.